# Expanding Reasoning Potential in Foundation Model by Learning Diverse Chains of Thought Patterns

**Xuemiao Zhang**[1,3*]    **Can Ren**[1,3*]   **Chengying Tu**[2,3*]
**Rongxiang Weng**[3†]    **Shuo Wang**[3]    **Hongfei Yan**[2,4†]    **Jingang Wang**[3]    **Xunliang Cai**[3]
[1] Peking University    [2] School of Computer Science, Peking University    [3] Meituan
[4] State Key Laboratory of Multimedia Information Processing, Peking University
{zhangxuemiao, yanhf}@pku.edu.cn  wengrongxiang@gmail.com
{tuchengying, 2401210098}@stu.pku.edu.cn
{wangshuo81, wangjingang02, caixunliang}@meituan.com

## Abstract

Recent progress in large reasoning models for challenging mathematical reasoning has been driven by reinforcement learning (RL). Incorporating long chain-of-thought (CoT) data during mid-training has also been shown to substantially improve reasoning depth. However, current approaches often utilize CoT data indiscriminately, leaving open the critical question of which data types most effectively enhance model reasoning capabilities. In this paper, we define the foundation model's *reasoning potential* for the first time as the inverse of the number of independent attempts required to correctly answer the question, which is strongly correlated with the final model performance. We then propose utilizing diverse data enriched with high-value reasoning patterns to expand the reasoning potential. Specifically, we abstract atomic reasoning patterns from CoT sequences, characterized by commonality and inductive capabilities, and use them to construct a core reference set enriched with valuable reasoning patterns. Furthermore, we propose a dual-granularity algorithm involving chains of reasoning patterns and token entropy, efficiently selecting high-value CoT data (CoTP) from the data pool that aligns with the core set, thereby training models to master reasoning effectively. Only 10B-token CoTP data enables the 85A6B Mixture-of-Experts (MoE) model to improve by **9.58%** on the challenging AIME 2024 and 2025, and to raise the upper bound of downstream RL performance by **7.81%**. [1]

## 1 Introduction

Recent progress in large reasoning models (LRMs) for challenging mathematical reasoning has largely been driven by post-training optimization, particularly via RL frameworks that improve problem-solving abilities through exploratory feedback (Zeng et al., 2025; Zheng et al., 2025; Schulman et al., 2017; Shao et al., 2024a). Empirical studies (Chen et al., 2025; Yue et al., 2025; Liu et al., 2025a; Zhao et al., 2025b; Wen et al., 2025) have revealed some critical insights that the parameter space of the foundation model inherently contains latent pathways for challenging reasoning and RL training effectively operationalizes the explicit manifestation of these implicit capabilities. In other words, *the reasoning capability learned in foundation models directly influences and limits the upper bounds of RL performance*, with certain open-source foundation models like Llama (Grattafiori et al., 2024) displaying unstable RL performance, emphasizing the urgent need for a thorough exploration of foundation model reasoning capabilities.

Recent studies suggest that blending question-answer (QA) data with chain-of-thought (CoT) during the mid-training stage, especially long-CoT samples, can markedly enhance the depth of reasoning

---

[*]Equal contribution.
[†]Corresponding author.

[1]The core code and dataset are available at `https://github.com/rc314159-creator/CoTP`.

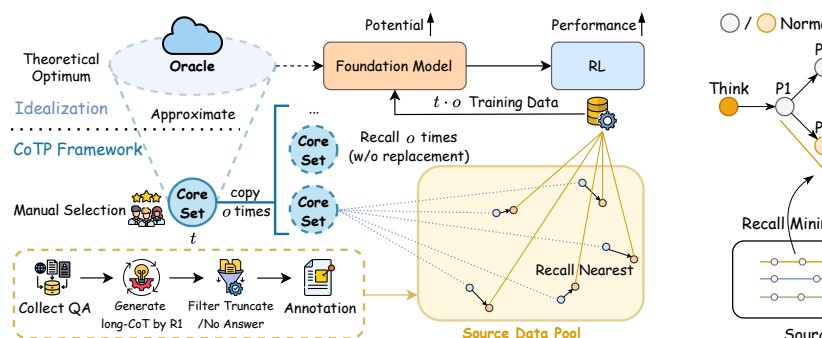

(a) The CoTP framework that selects long-CoT data similar to the core set based on the chains of reasoning patterns and token entropy.

(b) Graph of core set reasoning pattern chains.

Figure 1: Illustration of the CoTP framework. The left figure shows the overall process of the CoTP framework, while the right figure shows the graph of reasoning chains. The patterns with higher TF-IDF weights are important, while the remaining patterns are considered normal. The CoTP framework selects the minimum distance chain from the source data pool.

needed to tackle challenging problems (Wang et al., 2025b; Zhang et al., 2025b; Tu et al., 2025). Prevailing approaches primarily focus on augmenting datasets with challenging problems to generate long-CoT trajectories through *knowledge distillation*. However, these methods typically employ CoT data in a coarse-grained manner, lacking a thorough investigation into the inherent paradigmatic characteristics and deeper essence of reasoning within CoT sequences for nuanced and differentiated applications. Therefore, elucidating the reasoning paradigms that can significantly expand the reasoning potential of foundation models represents a pivotal research direction.

Building on these insights, our research focuses on how to effectively enrich reasoning patterns during mid-training to expand the reasoning potential of foundation models. We first theoretically define the reasoning potential of the foundation model and demonstrate that expanding this potential is equivalent to reducing the average number of reasoning attempts needed to correctly answer the question. We then abstract atomic reasoning patterns from CoT sequences, characterized by commonality and inductive capabilities, and use these to construct a core reference set enriched with valuable reasoning patterns. We propose a dual-granularity algorithm using weighted Dynamic Time Warping (DTW) to select long-CoT data exhibiting high-value reasoning patterns similar to those in the core set based on both the chains of reasoning patterns and token entropy.

Extensive experiments demonstrate that *mid-training on just 10B high-value reasoning data can significantly expand the reasoning potential of the 85A6B Mixture-of-Experts foundational model (MoE-6B-85B) and substantially raise the upper bounds of RL performance*. In summary, our contributions are as follows:

1. We theoretically define the reasoning potential of the foundation model for the first time as the inverse of the number of independent attempts required to correctly answer the question and abstract reasoning patterns that exhibit commonality and inductive capabilities from CoT data, guiding the construction of a high-value core reference set.

2. We propose a dual-granularity algorithm involving chains of reasoning patterns and token entropy, which efficiently selects CoT data with high-value reasoning patterns that aligns with the core reference set from the data pool, to effectively enrich reasoning patterns and expand the reasoning potential of foundation models.

3. We construct a long-CoT reasoning dataset, CoTP, which enables the 85A6B MoE model to maintain general performance while achieving a **9.58%** improvement on the challenging AIME 2024 and 2025 and raising the upper bound of downstream RL performance by **7.81%**.

## 2 CoTP: High-Value Reasoning Patterns for Efficient Reasoning

Our research aims to train the foundation model $\mathcal{M}_0$ to learn a wide variety of high-value reasoning patterns during the mid-training stage, thereby expanding the reasoning potential of $\mathcal{M}_0$ and incentivizing RL performance. We begin by theoretically defining and analyzing the reasoning potential of the foundation model. By abstracting atomic reasoning patterns from CoT sequences, we construct a core reference set enriched with valuable reasoning patterns to approximate the oracle reasoning data. As depicted in Figure 1a, our CoTP framework effectively selects long-CoT data similar to the core set from the source data pool using chains of reasoning patterns and token entropy, aided by a dual-granularity algorithm involving weighted DTW.

### 2.1 Problem Formulation

We theoretically define and analyze the model potential. Unlike conventional deterministic model evaluation, we assess the model potential by adopting the sampling mode inference multiple times to capture the stochastic nature of model performance. The model potential is then defined as follows:

**Definition 1 (Model Potential)** *For a given model $\mathcal{M}$ in sampling decoding mode and a question $q_i$, we define the model potential $\Phi(\mathcal{M}, q_i)$ as the probability that the model generates the correct answer for question $q_i$ when sampling from its output distribution:*

$$\Phi(\mathcal{M}, q_i) = \mathbb{P}[f_\mathcal{M}(q_i) = a_i^*], \tag{1}$$

*where $f_\mathcal{M}(q_i)$ denotes the sampled output and $a_i^*$ is the correct answer. The overall model potential $\Phi(\mathcal{M}, \mathcal{D}_{eval})$ is defined as the expected potential on an evaluation dataset $\mathcal{D}_{eval} = \{(q_i, a_i)\}_{i=1}^{N_{eval}}$:*

$$\Phi(\mathcal{M}, \mathcal{D}_{eval}) = \mathbb{E}_{(q,a)\sim\mathcal{D}_{eval}}[\Phi(\mathcal{M}, q)] = \frac{1}{N_{eval}} \sum_{i=1}^{N_{eval}} \Phi(\mathcal{M}, q_i). \tag{2}$$

There is a fundamental relationship between model potential and expected reasoning cost.

**Corollary 1** *Let $K_i$ denote the first-passage time for question $q_i$, representing the number of independent attempts required to solve $q_i$. Suppose each attempt is an independent Bernoulli trial with success probability $\Phi(\mathcal{M}, q_i)$, so that $K_i \sim \mathrm{Geom}(\Phi(\mathcal{M}, q_i))$. Then,*

$$\Phi(\mathcal{M}, q_i) = \frac{1}{\mathbb{E}[K_i]}. \tag{3}$$

*In other words, the model potential is the inverse of the expected first-passage time and a smaller $K_i$ indicates higher model potential.*

To achieve a lower $K$ for any given question, a promising approach is to extensively expose the model to a wide range of effective reasoning paradigms and enable it to internalize them. This insight motivates us to construct an ideal oracle dataset for training LLMs, which consists of samples that exhibit high reasoning efficiency, i.e., those with low-$K$ characteristics.

**Objective.** Formally, assume there exists an ideal oracle training dataset $\mathcal{D}_{oracle}^*$ that enables the foundation model to achieve maximal reasoning potential. Our goal is to select a training subset $\mathcal{D}_{train}^*$ containing $M$ samples from a given source dataset $\mathcal{D}_{source} = \{(q_j, c_j, a_j, \ell_j)\}_{j=1}^N$ to minimize the gap between the reasoning potential of the model trained on $\mathcal{D}_{train}^*$ and that trained on $\mathcal{D}_{oracle}^*$:

$$\mathcal{D}_{oracle}^* = \arg\max_{\mathcal{D}} [\Phi(\mathcal{M}_\mathcal{D}, \mathcal{D}_{eval}) - \Phi(\mathcal{M}_0, \mathcal{D}_{eval})] \tag{4}$$

$$\mathcal{D}_{train}^* = \arg\min_{\mathcal{D}\subset\mathcal{D}_{source}, |\mathcal{D}|=M} \left|\Phi(\mathcal{M}_\mathcal{D}, \mathcal{D}_{eval}) - \Phi(\mathcal{M}_{\mathcal{D}_{oracle}^*}, \mathcal{D}_{eval})\right|, \tag{5}$$

where $\mathcal{M}_\mathcal{D}$ denotes the model trained on dataset $\mathcal{D}$, $c_j$ and $a_j$ are the CoT and answer for question $q_j$, and $\ell_j$ denotes metadata labels such as subject and difficulty.

## 2.2 CORE SET APPROXIMATES ORACLE

Since it is difficult to determine $\mathcal{D}^*_{oracle}$, we propose approximating it using a carefully constructed reference core set consisting of CoT data rich in diverse high-value reasoning patterns, as shown in Figure 1b. This core set will guide the selection of $\mathcal{D}_{train}$ from $\mathcal{D}_{source}$.

We propose capturing the reasoning properties of each CoT sample at two granularities: using pattern chains to capture highly abstract reasoning paradigms (Chen et al., 2025), and using entropy chains to capture token features with high reasoning gain (Wang et al., 2025a; Cui et al., 2025).

**Definition 2 (Reasoning Pattern and Pattern Chain)** *A reasoning pattern $\rho$ is an atomic cognitive operation representing a fundamental reasoning step applicable across diverse problem domains (see Figure 5). And a pattern chain $\mathcal{C} = [\rho_1, \rho_2, \ldots, \rho_n]$ is an ordered sequence of reasoning patterns extracted from a CoT sequence (see Figure 4).*

We employ Deepseek-V3 (Liu et al., 2024) to annotate the reasoning pattern chain for each CoT sequence (detailed in Appendix E). The entropy of each token in the CoT sequence is computed as $h_t = -\sum_{v \in \mathcal{V}} p_t(v) \log p_t(v)$, where $p_t(v)$ denotes the probability of token $v$ at position $t$. Consequently, the entropy chain is represented as $\mathcal{H} = [h_1, h_2, \ldots, h_T]$. Illustrative examples of annotated reasoning pattern chains and entropy chains for CoT data are provided in Appendix G.3.

Formally, we define extraction functions $\xi : \mathcal{C} \rightarrow \mathcal{P}^*$ and $\eta : \mathcal{C} \rightarrow \mathbb{R}^T$, mapping each CoT sequence $c$ to its pattern chain $\xi(c)$ and entropy chain $\eta(c)$. (1) To construct the core set, we first filter questions from the source dataset $\mathcal{D}_{source}$, annotated with difficulty levels and problem types. Questions are randomly sampled to match the expected distribution. We then employ multiple strong reasoning models to independently generate answers, using majority voting to determine the correct ones, and filter out low-quality data, such as unsolvable questions, yielding $\mathcal{Q} = \{q_1, \ldots, q_{|\mathcal{Q}|}\}$. (2) For each $q_i \in \mathcal{Q}$, we employ a strong reasoning model to generate $r$ CoT sequences denoted as $\{c_{i,j}\}_{j=1}^r$. We extract the pattern chains $\xi(c_{i,j})$ and assess the importance of each pattern $\rho_k$ for the question $q_i$ using the TF-IDF weighting scheme:

$$\omega(\rho_k \mid q_i, \mathcal{Q}) = \text{TF}(\rho_k, q_i) \times \text{IDF}(\rho_k, \mathcal{Q}), \tag{6}$$

where the calculation details for TF and IDF are provided in Appendix B.1. (3) From the remaining CoT sequences with correct answers, according to the importance scores of different patterns, we manually select those exhibiting distinctive, high-importance patterns to construct the core set $\mathcal{C}_{core} = \{(q_i, c_i, a_i, \ell_i)\}_{i=1}^t$ (see Figure 1b), where each instance is accompanied by its pattern importance weights $\Omega_i = \{\omega(\rho_k \mid q_i, \mathcal{Q})\}$ for $\rho_k \in \xi(c_i)$.

## 2.3 SELECT TRAINING DATA WITH HIGH-VALUE REASONING PATTERNS

Given $\mathcal{C}_{core} = \{(q_i^c, c_i^c, a_i^c, \ell_i^c)\}_{i=1}^t$, we construct a training dataset $\mathcal{D}_{train}$ of size $T$ by selecting instances with similar CoT sequences from the source dataset $\mathcal{D}_{source} = \{(q_j^s, c_j^s, a_j^s, \ell_j^s)\}_{j=1}^N$. For analytical convenience, we set $T = t \cdot o$, where $T < N$, ensuring each core instance is associated with $o$ source instances. We formulate this as an assignment problem with capacity constraints. Let $\mathbf{D} \in \mathbb{R}^{t \times N}$, where $D_{ij}$ denotes the distance between CoT sequences $c_i^c$ and $c_j^s$. We seek to optimize a binary matrix $\mathbf{S} \in \{0,1\}^{t \times N}$, where $S_{ij} = 1$ signifies the assignment of source instance $j$ to core instance $i$. The objective is given by:

$$\min_{\mathbf{S}} \quad \sum_{i=1}^t \sum_{j=1}^N D_{ij} S_{ij}$$

$$\text{s.t.} \quad \sum_{j=1}^N S_{ij} = o, \forall i \in [1, t]; \quad \sum_{i=1}^t S_{ij} \leq 1, \forall j \in [1, N]; \quad S_{ij} \in \{0, 1\} \tag{7}$$

The distance $D_{ij}$ is the weighted sum of pattern chain and entropy chain distances, with $\lambda \in [0, 1]$:

$$D_{ij} = \lambda\, d_{\text{pattern}}(\xi(c_i^c), \xi(c_j^s)) + (1 - \lambda)\, d_{\text{entropy}}(\eta(c_i^c), \eta(c_j^s)). \tag{8}$$

For distance computation, we employ DTW for both dimensions:

$$d(x, y) = \text{WeightedDTW}(x, y, w, \delta). \tag{9}$$

We set the parameters as follows: for the pattern chain distance, $x = \xi(c_i^c)$, $y = \xi(c_j^s)$, $w = \Omega_i$, and $\delta = d_{\text{ngram}}$ (see Figure 4); for the entropy chain distance, $x = \eta(c_i^c)$, $y = \eta(c_j^s)$, $w = \mathbf{1}$, and $\delta = d_{\text{abs}}$. The distance computation and WeightedDTW are detailed in Algorithms 2 and 3.

To efficiently solve this optimization problem, we reformulate it as a standard linear assignment problem by replicating each core instance $o$ times, resulting in an expanded cost matrix of size $t \cdot o \times N$ with entries $D_{ij}$. The optimal assignment is then obtained using the Hungarian algorithm (Kuhn, 1955; Mills-Tettey et al., 2007). This transformation guarantees optimality, as each source instance is assigned to at most one core, and the replication ensures that each core receives exactly $o$ assignments (see Appendix B.2 for proof). The overall data selection procedure of our CoTP framework is summarized in Algorithm 1. Note that it is domain-agnostic and theoretically applicable to any scenario decomposable into atomic reasoning patterns (see Appendix F.3 for pattern visualization in STEM domains).

---

**Algorithm 1** Hungarian Data Selection Algorithm

**Input:** Source dataset $\mathcal{D}_{pool}$; Core Set $\mathcal{C}_{core}$; Weight parameter $\lambda$
**Output:** Final selected dataset $\mathcal{D}_{select}$
1: Initialize $\mathcal{D}_{select} \leftarrow \emptyset$, $\mathbf{D} \in \mathbb{R}^{|\mathcal{D}_{pool}| \times |\mathcal{C}_{core}|}$
2: **for** each $(q_j^s, c_j^s, a_j^s, \boldsymbol{\ell}_j^s) \in \mathcal{D}_{pool}$, $i = 1$ **to** $m$ **do**
3:     $d_{\text{pattern}} \leftarrow \text{WeightedDTW}(\xi(c_j^s), \xi(c_i^c), \Omega_i, d_{\text{ngram}})$ // refer to Algorithm 3 and 2
4:     $d_{\text{entropy}} \leftarrow \text{WeightedDTW}(\eta(c_j^s), \eta(c_i^c), \mathbf{1}, d_{\text{abs}})$ // refer to Algorithm 3
5:     $\mathbf{D}[j, i] \leftarrow \lambda \cdot d_{\text{pattern}} + (1 - \lambda) \cdot d_{\text{entropy}}$
6: assignment$_j \leftarrow \text{Hungarian}(\mathbf{D}_j)$ // Input: cost matrix; Output: optimal assignment
7: $\mathcal{D}_{select} \leftarrow \{(q_s^s, c_s^s, a_s^s, \boldsymbol{\ell}_s^s) : s \in \text{selected indices from assignment}_j\}$
8: **return** $\mathcal{D}_{select}$

---

# 3 EXPERIMENTS

## 3.1 EXPERIMENTAL SETUP

**Data Construction.** To construct a high-quality reasoning data pool, we integrate diverse mathematical QA datasets as follows and conduct rigorous n-gram deduplication: (1) OpenR1-Math-220k (Hugging Face, 2025) (OpenR1-Math) comprises 220k math problems, each expanded with two to four reasoning traces generated by DeepSeek-R1 (Guo et al., 2025); (2) AM-DeepSeek-R1-Distilled (Zhao et al., 2025a) (AM-Distilled) focuses on general reasoning tasks, with detailed thinking traces; (3) BoostQA (Zhang et al., 2025b) consists of large-scale QA pairs of different difficulty levels. We specifically select high-difficulty H4/H5-level questions from BoostQA and employ DeepSeek-R1 to generate long reasoning CoT, maintaining consistency with OpenR1-Math and AM-Distilled, with a maximum output length set to 32k tokens. In industrial production, the use of multiple reasoning models is encouraged to enrich the variety of reasoning patterns. We construct the LongCoTPool (see Figure 1a) by excluding truncated or unanswered data, annotating chains of reasoning patterns, and ensuring no overlap with the core set. Following Shao et al. (2024b), we employ exact 10-gram matching and embedding-based similarity filtering to mitigate contamination of questions and answers originating from benchmarks.

**Training Details.** We conduct mid-training experiments using the 85A6B MoE (Jiang et al., 2024) foundation model, which is pre-trained on 14T-token corpora. During the mid-training stage, the model decays on a mixture of 30B-token specialized experimental reasoning data and general-domain data, KnowEdu (Zhang et al., 2025b) at a 1:2 ratio. The reasoning data follows a structured format of $\{question\}\backslash n\{cot\_answer\}$ with final answers encapsulated in $\backslash boxed\{\}$. In scaling experiments, the data volume is expanded to 60B tokens while maintaining the same data blend ratio. We further conduct SFT using the same collected dataset of 60k long-CoT entries to enhance the models' capability to generate long-CoT sequences. *This step is crucial for ensuring fair comparisons across models, as it prevents underestimation of models that initially lack the ability to produce long-CoT outputs* (see analysis in Appendix D.2). Moreover, the SFT stage facilitates smooth transitions to RL by establishing a robust foundation in CoT reasoning for effective rollouts. Importantly, we apply the same SFT data across all models to eliminate any variances and uphold consistency

Table 1: General performance and average pass@1 accuracy (%) of models. The best and second-best are in bold and underlined, respectively. Abbreviations: Beyond = BeyondAIME.

| Dataset | General | AIME2025 | AIME2024 | HMMT2025 | Beyond | MATH500 | AVG. |
|---|---|---|---|---|---|---|---|
| KnowEdu | 64.39 | 0.33±0.35 | 1.22±0.68 | 5.10±1.40 | 0.00±0.00 | 45.80±4.37 | 10.49±1.36 |
| BoostQA | 63.29 | 0.52±0.46 | 1.46±0.76 | 4.06±1.25 | 0.40±0.39 | 54.00±4.37 | 12.09±1.45 |
| JiuZhang3.0 | 64.32 | 0.83±0.57 | 1.25±0.71 | 2.92±1.06 | 0.10±0.20 | 56.00±4.36 | 12.22±1.38 |
| MegaMathQA | 64.79 | 0.21±0.29 | 2.19±0.93 | 4.17±1.25 | 0.40±0.39 | 51.80±4.38 | 11.75±1.45 |
| OMInstruct-2 | 66.24 | 3.44±1.15 | 8.02±1.73 | 5.42±1.44 | 1.20±0.67 | 72.20±3.93 | 18.06±1.78 |
| OpenR1-Math | 66.58 | 23.96±2.74 | 29.69±2.92 | 16.04±2.34 | 9.10±1.79 | 87.80±2.87 | 33.32±2.53 |
| AM-Distilled | 67.97 | 23.12±2.70 | 25.52±2.79 | 18.02±2.46 | 8.30±1.72 | 87.20±2.93 | 32.43±2.52 |
| LongCoTPool | 65.95 | 21.89±2.46 | 24.90±2.85 | 15.63±2.31 | 7.90±1.72 | 85.40±3.10 | 31.14±2.49 |
| CoTP (Ours) | 66.08 | **28.02**±2.88 | **37.92**±3.09 | **20.73**±2.58 | **10.20**±1.88 | **90.80**±2.54 | **37.53**±2.59 |

in evaluation conditions. Furthermore, we conduct experiments to verify that the expanded reasoning potential of the foundation model can enhance downstream RL performance. Specifically, we apply the same experimental settings to foundation models trained on different datasets to evaluate their subsequent RL performance, where the RL algorithm adopts GSPO (Zheng et al., 2025). The detailed setup is provided in Appendix C.1.

**Evaluation.** We conduct a comprehensive evaluation of mid-trained models to assess their capability to sustain general performance after exposure to reasoning-intensive data, including benchmarks such as MMLU (Hendrycks et al., 2021a), CMMLU (Li et al., 2024), C-Eval (Huang et al., 2023), WinoGrande (Sakaguchi et al., 2021), HellaSwag (Zellers et al., 2019), ARC-C (Clark et al., 2018), BIG-Bench (Suzgun et al., 2023) and DROP (Dua et al., 2019). Furthermore, we conduct assessments of the SFT models on downstream challenging mathematical reasoning tasks to examine the reasoning potential of the foundation models. The benchmarks include AIME 2025 & 2024 (MAA), HMMT 2025 (HMMT), BeyondAIME (ByteDance-Seed, 2025), and MATH500 (Hendrycks et al., 2021b), each repeated multiple times for statistical robustness (detailed in Appendix C.2). Pass@k (Chen et al., 2021) curves are drawn to provide a detailed visualization of model performance dynamics with increasing attempts $k$. For the RL stage, the same benchmarks are employed to evaluate the enhancements in reasoning capabilities.

**Baselines.** Baselines can be divided into two categories, which differ in mid-training data with the same other settings. The first paradigm evaluates the 30B-token general corpus KnowEdu (Zhang et al., 2025b), a high-quality knowledge-rich educational dataset. The second paradigm assesses the QA blend following the same 1:2 blend ratio between QA data and KnowEdu. BoostQA (Zhang et al., 2025b) contains QA data without CoT sequences. The short-CoT QA datasets include JiuZhang3.0 (Zhou et al., 2024), MegaMathQA, a QA subset from MegaMath-Synthetic (Zhou et al., 2025), and OpenMathInstruct-2 (Toshniwal et al., 2024) (OMInstruct-2). The long-CoT QA datasets include OpenR1-Math, AM-Distilled, and our curated data pool, LongCoTPool. (detailed in Table 5)

## 3.2 MAIN RESULTS

**Only 10B high-value reasoning data selected by CoTP can significantly improve multiple challenging mathematical reasoning tasks by an average of 6.39%, reaching SOTA results and substantially raising the upper bound of RL performance.** As depicted in Table 1, CoTP not only sustains performance across general benchmarks, but also surpasses baselines on multiple challenging mathematical reasoning benchmarks, with average gains of 4.21% compared to OpenR1-Math and 5.10% compared to AM-Distilled. More detailed discussions on general performance are shown in Appendix D.1. It offers an average improvement of 6.39% over LongCoTPool, with a remarkable **9.58%** enhancement on AIME 2025 & 2024. Furthermore, LongCoTPool, as a mixed data pool, shows a slight decline in average performance relative to OpenR1-Math and AM-Distilled, indicating that a simple mixing of reasoning data might be insufficient to expand the model's reasoning potential. In contrast, CoTP, curated from LongCoTPool, achieves optimal performance, underscoring the importance of selecting high-value reasoning data and the advantages of our CoTP framework.

Table 2: Comparison of the performance of models mid-trained on different datasets.

| Dataset | AIME 2025 | | AIME 2024 | | HMMT 2025 | | BeyondAIME | | MATH500 | | AVG. | |
|---|---|---|---|---|---|---|---|---|---|---|---|---|
| | SFT | RL | SFT | RL | SFT | RL | SFT | RL | SFT | RL | SFT | RL |
| KnowEdu | 0.33 | 0.31 | 1.22 | 0.83 | 5.10 | 1.25 | 0.00 | 0.00 | 45.80 | 44.60 | 10.49 | 9.40 |
| LongCoTPool | 21.89 | 31.88 | 24.90 | 44.38 | 15.63 | 35.31 | 7.90 | 16.40 | 85.40 | 90.20 | 31.14 | 43.63 |
| CoTP (Ours) | 28.02 | 40.81 | 37.92 | 58.65 | 20.73 | 41.35 | 10.20 | 23.20 | 90.80 | 93.20 | 37.53 | 51.44 |

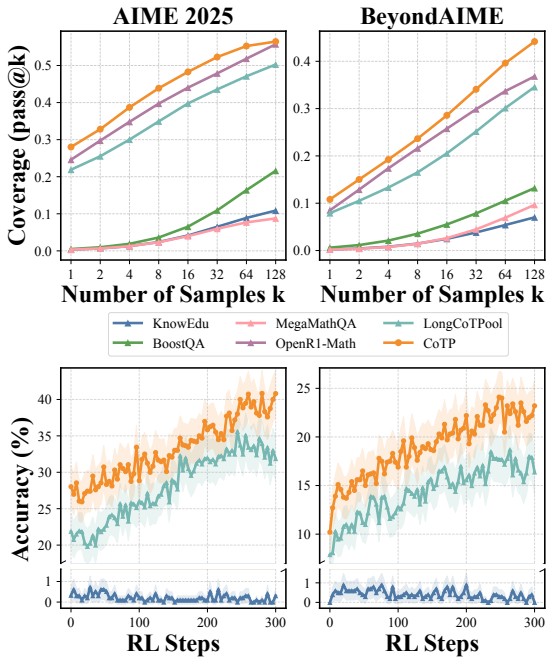

Figure 2: The comparison of pass@k and RL performance across different datasets.

As depicted in Figure 2, in the pass@k evaluation, models mid-trained on CoT datasets, particularly CoTP, show consistent and progressive improvement as the number of attempts $k$ increases, along with superior performance as the RL training steps increase, consistent with findings from OctoThinker (Wang et al., 2025b). This growth highlights *the expanded model potential during the mid-training stage (see Table 2), which can substantially raise the upper bound of RL performance.* This demonstrates that the improvements introduced during mid-training can be effectively carried over to RL, rather than being prematurely acquired during mid-training and thereby diminishing the distinct benefits typically observed in RL, which would otherwise result in no significant difference in final model performance. Our CoTP dataset stands out by exhibiting superior mathematical reasoning capabilities, with an average improvement of **7.81%** over LongCoTPool and 42.04% over KnowEdu. This highlights its efficacy for expanding challenging mathematical reasoning potential, thereby incentivizing RL performance.

**CoTP exhibits exceptional scalability in challenging mathematical reasoning tasks.** As illustrated in Figure 3 (detailed in Appendix D.3), when scaled to 60B tokens—with additional data incorporated by relaxing the similarity threshold—the model sustains its upward performance trajectory, achieving a 4.72% average improvement on AIME 2025 & 2024 compared to the results at 30B tokens. This suggests that CoTP effectively expands the model's potential in challenging mathematical reasoning tasks even at larger data volumes while maintaining general performance, demonstrating its continuous effectiveness in recalling high-value reasoning data.

## 3.3 ABLATIONS

We conduct ablation studies to analyze the impact of various components in the CoTP framework. For efficiency and rapid validation, these ablation experiments are performed on the 12B-token QA blend, maintaining the same 1:2 ratio between QA data and KnowEdu and and using the same experimental settings as in the main experiments. The results are shown in Table 3.

**Entropy-based selection enhances the quality of reasoning data accessed.** By employing a reference model to perform an offline evaluation of the information content of each token within the CoT reasoning data (Wang et al., 2025a), entropy-based selection enables a more fine-grained capture of token-level reasoning paradigms. The utility of this approach is further illustrated by the

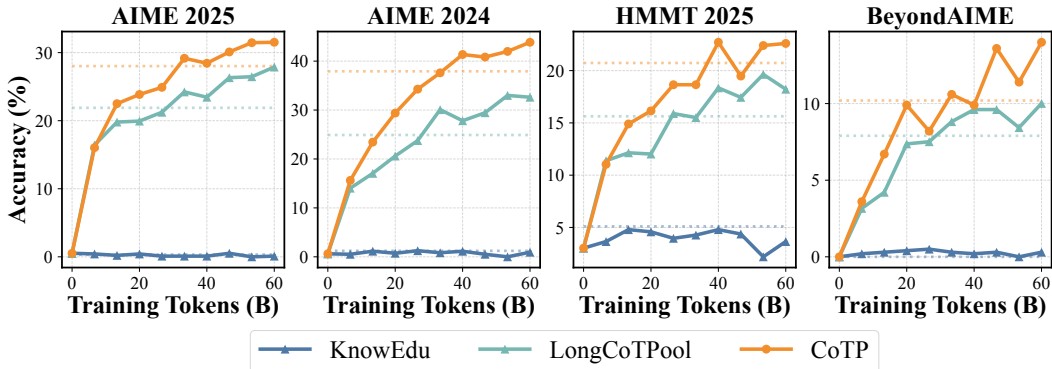

Figure 3: Scalability of data volume examining the SFT performance of models mid-trained on datasets of varying volumes. The dashed lines represent the performance of each dataset configured under the 30B-token setting.

Table 3: Ablation results. CoTP uses *n=1 or 2* for n-gram setting, $\lambda$=0.8 for entropy. And w/o entropy denotes $\lambda$=1.

| Dataset | AIME 2025 | AIME 2024 | HMMT 2025 | BeyondAIME | MATH500 | AVG. |
|---|---|---|---|---|---|---|
| **CoTP** | 21.84±2.68 | 25.76±2.70 | 13.90±2.27 | 6.70±1.55 | 85.20±3.25 | 30.68±2.49 |
| **w/o entropy** | 21.46±2.63 | 25.63±2.78 | 11.88±2.06 | 6.30±1.51 | 84.40±3.18 | 29.93±2.43 |
| **n-gram n=2** | 18.33±2.48 | 25.42±2.77 | 11.56±2.04 | 6.50±1.53 | 83.80±3.23 | 29.12±2.41 |
| **w/o importance** | 19.69±2.54 | 24.17±2.72 | 13.65±2.19 | 6.60±1.54 | 84.20±3.20 | 29.66±2.44 |

entropy visualization in Appendix G.2, which provides deeper insights into the structural nuances captured. This enhanced quality correlates with improved model performance, underscoring the pivotal role of entropy-based selection in fostering advanced reasoning capabilities.

**For n-gram pattern similarity, *n=1 or 2* yields superior outcomes, providing a more balanced evaluation of pattern similarity.** This configuration integrates the broader contextual encapsulation of n=1 with the detailed specificity of n=2, offering a comprehensive representation of pattern alignments. Notably, unlike English, each character in Chinese carries intrinsic semantic meaning and we demonstrate that Chinese pattern representation exhibits clear distinctions between unrelated pattern pairs under the character-level n-gram method (detailed in Appendix F.2), which makes it more suitable for calculating distances between pattern entries in Algorithm 2. Consequently, we employ Chinese for annotating reasoning pattern chains.

**Importance scores play a critical role in the CoTP framework.** The model lacking importance weighting shows poorer performance, underscoring the significance of pattern importance scores. It highlights the distinction of normal and important reasoning patterns, due to their different contribution to reasoning potential.

## 4 ANALYSIS

**Our CoTP-trained model demonstrates an enhanced mastery over a broader array of correct and key reasoning patterns.** To validate whether CoTP enables models to learn key reasoning patterns closer to the ideal ones, we conduct a comparative analysis of reasoning patterns exhibited by models trained on various datasets, taking AIME 2025 & 2024 as examples. Specifically, we consider the model trained on the oracle set as the ideal model, whose generated reasoning traces encompass comprehensive patterns. Our core set serves as an approximation of this oracle. Through the data selection algorithm, we select training data that aligns with the core set distribution. In

Figure 4: Illustration of DTW alignment analysis on the pattern chain similarity matrix.

(a) Reasoning patterns with high importance scores.

(b) Reasoning patterns with low importance scores.

Figure 5: Examples of reasoning patterns with different levels of importance.

this analysis, we use the strong reasoning model DeepSeek-R1 as a proxy for the ideal model. Correct patterns are defined as those previously appearing in correct reasoning pattern chains.

As illustrated in Table 4 (detailed in Appendix F.1), CoTP not only increases the number of correct patterns it utilizes but also shows an enhanced overlap with the key patterns mastered by DeepSeek-R1, with an increase of 7.25% compared to LongCoTPool. This advancement suggests that our training methodology effectively improves the model's capability to solve challenging reasoning tasks by systematically integrating and aligning with the reasoning paradigms established by highperformance models like DeepSeek-R1.

Table 4: A comparative analysis of reasoning patterns on AIME 2025 & 2024.

| Dataset | Patterns | Correct | Key |
|---|---|---|---|
| KnowEdu | 4296 | 147 | 116 (1.79%) |
| LongCoTPool | 3407 | 2246 | 1345 (20.73%) |
| CoTP | 3970 | 3226 | 1815 (27.98%) |

**Our CoTP-trained model shows a notable convergence to the reasoning paradigms of the reference model, evidenced by a reduction in pattern chain distances.** By conducting a comprehensive pattern distance analysis of models before and after mid-training against DeepSeek-R1 on the core set, we quantify these alignment enhancements. As depicted in Figure 6a, we employ the DTW algorithm (see Figure 4) to calculate the reasoning pattern chain distances between the CoT responses from both the model instances before and after midtraining and those from DeepSeek-R1 within each question, then compute the Wasserstein distance between these cross-model distance distributions for each question, and finally average the results across all questions. The average distance between the model before mid-training and DeepSeek-R1 is 0.51, whereas our CoTP-trained model achieves a reduced distance of 0.35, marking a 31.4% improvement. Notably, CoTP acquires sophisticated reasoning patterns (see Figure 5a). These advanced reasoning patterns reflect expertlevel problem-solving approaches and underscore that CoTP effectively facilitates the acquisition of sophisticated reasoning capabilities critical for solving challenging mathematical problems.

**The composition of CoTP closely matches the distribution of the core set.** As shown in Figure 6b, the KL divergence between the problem type distributions relative to the core set decreases from 0.18 (LongCoTPool) to 0.04 (CoTP), demonstrating improved alignment. Furthermore, as illustrated in Figure 6c, the token length distribution of CoTP aligns more closely with the core set, which generally features longer token lengths, than with LongCoTPool. This alignment highlights the effectiveness of our CoTP framework in targeting the core set distribution.

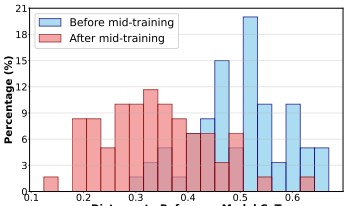
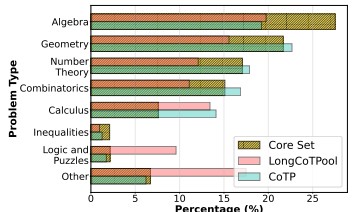
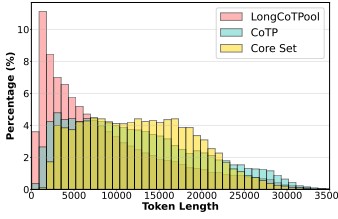

(a) Reasoning chain distance distributions relative to DeepSeek-R1.

(b) Distribution of problem types in different datasets.

(c) Distribution of token lengths in different datasets.

Figure 6: Distribution analysis.

**Effective reasoning patterns not only guide the steps of chain-of-thought, but also facilitate deeper introspective processes.** We investigate the relationship between reasoning patterns and reflection and examples in Appendix G.1 illustrate how reflection (Shah et al., 2025) can be seamlessly integrated into established reasoning pattern chains, showcasing the multifaceted characteristics of introspective processes. Our detailed examination reveals that pattern chains inherently encompass reflective characteristics. Reflection may be encapsulated within discrete patterns, providing precise cognitive insights, or may appear as recursive loops in pattern chains, indicating iterative thought refinement. This analysis to investigating reasoning and reflection indicates their intertwined nature.

## 5 RELATED WORK

Endowing LLMs with reasoning capabilities remains challenging, particularly in the realm of complex mathematical reasoning. Recent advancements in RL algorithm optimization strategies have significantly enhanced the performance of LLMs on downstream challenging mathematical reasoning tasks (Chen et al., 2025; Yue et al., 2025; Liu et al., 2025a). These improvements are largely driven by exploration-feedback mechanisms that systematically elevate the problem-solving skills of these models. Additionally, they have unveiled the inherent constraints on RL performance imposed by the reasoning capability reserves of the foundation models. Studies investigating the interaction between the foundation models and RL performance (Gandhi et al., 2025; Liu et al., 2025b; Wang et al., 2025b), such as those related to cognitive behaviors that facilitate self-improvement and insights from training paradigms like R1-Zero, have demonstrated that different foundation models exhibit diverse initial reasoning behaviors. These behaviors critically influence their ability for self-improvement, thereby manifesting varied RL scaling characteristics.

Among these studies, OctoThinker (Wang et al., 2025b) has validated that integrating CoT QA data, particularly long-CoT samples, during the mid-training stage significantly enhances the reasoning capabilities required for complex problem-solving. This approach offers a promising avenue for enhancing the reasoning capabilities of foundation models. Notable large-scale open-source long-CoT QA datasets include OpenR1-Math-220k (Hugging Face, 2025) and AM-DeepSeek-R1-Distilled (Zhao et al., 2025a), with their long-CoT responses predominantly generated by DeepSeek-R1. Such developments underscore the importance of strategically aligning mid-training data with downstream tasks as a potent strategy for expanding the reasoning proficiency of LLMs.

## 6 CONCLUSION

In conclusion, our proposed CoTP framework presents a significant advancement in expanding LLM reasoning potential. By defining reasoning potential as the inverse of the number of independent attempts required to correctly answer the question, we have proposed a framework that constructs high-value reasoning data. Our dual-granularity algorithm, which leverages reasoning patterns and token entropy, efficiently selects valuable CoT data similar to the core reference set, thereby enriching reasoning patterns in foundation models. Through the construction of the CoTP dataset, we have enabled the 85A6B MoE foundational model to achieve a **9.58%** improvement on the challenging AIME 2025 & 2024 and to raise the upper bound of downstream RL performance by **7.81%**.

## 7 ETHICS STATEMENT

Our research adheres to the ICLR Code of Ethics. We have carefully read the ethical guidelines and ensured that our research does not present ethical concerns.

## 8 REPRODUCIBILITY STATEMENT

We provide comprehensive details throughout various sections, including the main paper, appendix, and supplementary materials, to ensure reproducibility. We ensure all experiments and analyses are clearly described, with additional details provided in the appendix. The source codes of the algorithm are provided in the supplementary materials. Explanations and complete proofs of the theoretical results are also included in the appendix. Details about the experiments, including the datasets and the computing infrastructure, are provided in both the appendix and the supplementary materials.

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

## A  THE USE OF LARGE LANGUAGE MODELS (LLMS)

We utilize LLMs solely to refine our manuscript. Specifically, LLMs are employed to identify and correct errors in spelling, grammar, punctuation, and formatting at the sentence level.

## B  MATHEMATICAL FORMULATIONS AND ALGORITHMS

### B.1  PATTERN IMPORTANCE CALCULATION VIA TF-IDF

We employ a TF-IDF weighting scheme to quantify the importance of reasoning patterns, balancing pattern frequency within individual questions against their global rarity across the dataset.

The importance of pattern $\rho_k$ for question $q_i$ is calculated using a TF-IDF weighting scheme:

$$
\begin{aligned}
\omega(\rho_k \mid q_i, \mathcal{Q}) &= \text{TF}(\rho_k, q_i) \times \text{IDF}(\rho_k, \mathcal{Q}) \\
\text{TF}(\rho_k, q_i) &= \frac{\text{count}(\rho_k \text{ in } \bigcup_{j=1}^{r} \xi(c_{i,j}))}{\sum_{\rho} \text{count}(\rho \text{ in } \bigcup_{j=1}^{r} \xi(c_{i,j}))} \\
\text{IDF}(\rho_k, \mathcal{Q}) &= \log\left(\frac{|\mathcal{Q}|}{|\{q_i : \rho_k \in \bigcup_{j=1}^{r} \xi(c_{i,j}), q_i \in \mathcal{Q}\}|}\right)
\end{aligned}
\tag{10}
$$

where $\xi(c_{i,j})$ represents the pattern chain extracted from the $j$-th CoT data of $q_i$.

### B.2  EQUIVALENCE TO PERFECT BIPARTITE MATCHING

We establish the theoretical foundation for our assignment optimization by proving the equivalence between the capacity-constrained assignment problem and the perfect bipartite matching problem.

**Theorem 2** *The capacity-constrained assignment problem in Eq. (7) is equivalent to a minimum weight perfect bipartite matching problem.*

**Proof.** We construct a balanced bipartite graph $G' = (U' \cup V', E')$ where both partitions have size $N$. The left partition $U' = \{u_{i,k} : i \in [1, t], k \in [1, o]\} \cup \{u_{\text{dummy},\ell} : \ell \in [1, N - t \cdot o]\}$ contains $t \cdot o$ replica nodes (each core instance $i$ replicated $o$ times) plus $(N - t \cdot o)$ dummy nodes. The right partition $V' = \{v_j : j \in [1, N]\}$ contains the original $N$ source instances.

Edge weights are defined as:

$$
w(u_{i,k}, v_j) = D_{ij}, \quad \forall i \in [1, t], k \in [1, o], j \in [1, N] \tag{11}
$$
$$
w(u_{\text{dummy},\ell}, v_j) = M, \quad \forall \ell \in [1, N - t \cdot o], j \in [1, N] \tag{12}
$$

where $M > \max_{i,j} D_{ij}$ is a sufficiently large constant.

The transformed problem seeks a minimum weight perfect matching in $G'$:

$$
\begin{aligned}
\min_{\mathbf{X}} \quad & \sum_{i=1}^{t} \sum_{k=1}^{o} \sum_{j=1}^{N} D_{ij} X_{i,k,j} + \sum_{\ell=1}^{N-t\cdot o} \sum_{j=1}^{N} M \cdot X_{\text{dummy},\ell,j} \\
\text{s.t.} \quad & \sum_{j=1}^{N} X_{u,j} = 1, \quad \forall u \in U' \\
& \sum_{u \in U'} X_{u,j} = 1, \quad \forall j \in [1, N] \\
& X_{u,j} \in \{0, 1\}
\end{aligned}
\tag{13}
$$

We establish equivalence by showing that feasible solutions correspond bijectively with identical objective values.

Given any feasible solution $\mathbf{S}$ to the original problem, we construct a perfect matching $\mathbf{X}$ as follows. For each core $i$, let $J_i = \{j : S_{ij} = 1\}$ be its assigned sources with $|J_i| = o$. Order $J_i$ arbitrarily

as $\{j_1^{(i)}, \ldots, j_o^{(i)}\}$ and set $X_{i,k,j_k^{(i)}} = 1$ for $k \in [1, o]$. This matches exactly $t \cdot o$ sources to replica nodes. The remaining $(N - t \cdot o)$ unmatched sources are matched to dummy nodes with cost $M$. The objective value is:

$$\sum_{i,k,j} D_{ij} X_{i,k,j} + (N - t \cdot o) \cdot M = \sum_{i,j} D_{ij} S_{ij} + (N - t \cdot o) \cdot M \tag{14}$$

Conversely, given any minimum weight perfect matching $\mathbf{X}$ in $G'$, we construct $\mathbf{S}$ by setting $S_{ij} = 1$ if there exists $k$ such that $X_{i,k,j} = 1$ for replica node $u_{i,k}$, and $S_{ij} = 0$ otherwise. Since $M > \max_{i,j} D_{ij}$, any optimal perfect matching will minimize the use of dummy nodes. Specifically, exactly $(N - t \cdot o)$ dummy nodes must be matched (due to perfect matching constraints), and the optimal solution will match all $t \cdot o$ replica nodes to distinct sources to minimize total cost. Therefore, each core $i$ has exactly $o$ matched replicas, giving $\sum_j S_{ij} = o$, and each source is matched to at most one replica, giving $\sum_i S_{ij} \leq 1$. The objective values satisfy:

$$\sum_{i,j} D_{ij} S_{ij} = \sum_{i,k,j} D_{ij} X_{i,k,j} \tag{15}$$

This bijective correspondence between feasible solutions with proportional objective values establishes that optimal solutions of both problems correspond exactly. ∎

### B.3 ALGORITHM

We implement two complementary distance metrics for measuring similarity between reasoning patterns and pattern chains: character n-gram cosine distance for lexical similarity (detailed in Algorithm 2) and weighted dynamic time warping for structural alignment (detailed in Algorithm 3).

---

**Algorithm 2** Character N-gram Cosine Distance

---

**Input:** String $a$; String $b$; Maximum n-gram length $n$
**Output:** N-gram cosine distance $d_{\text{ngram}}(a, b)$

1: $a \leftarrow \text{normalize}(a)$; $b \leftarrow \text{normalize}(b)$
2: Initialize frequency maps $F_a \leftarrow \{\}$, $F_b \leftarrow \{\}$
3: **for** $k = 1$ **to** $n$ **do**:
4:     **for** $i = 1$ **to** $|a| - k + 1$ **do**: $F_a[a[i : i + k - 1]] \leftarrow F_a[a[i : i + k - 1]] + 1$
5:     **for** $i = 1$ **to** $|b| - k + 1$ **do**: $F_b[b[i : i + k - 1]] \leftarrow F_b[b[i : i + k - 1]] + 1$
6: $\text{dot} \leftarrow \sum_g F_a[g] \cdot F_b[g]$
7: $\|F_a\|^2 \leftarrow \sum_g F_a[g]^2$; $\|F_b\|^2 \leftarrow \sum_g F_b[g]^2$
8: **if** $\|F_a\|^2 = 0$ **or** $\|F_b\|^2 = 0$ **then return** $0.0$
9: **return** $1.0 - \frac{\text{dot}}{\sqrt{\|F_a\|^2 \cdot \|F_b\|^2}}$

---

## C EXPERIMENTAL SETUP

### C.1 TRAINING DETAILS

We use H800 to mid-train our pre-trained 85A6B Mixture-of-Experts foundation model, which is pre-trained on 14T-token corpora. The model decays on 30B tokens of specialized experimental reasoning data and general-domain data, KnowEdu (Zhang et al., 2025b;a) at a 1:2 ratio, using the WSD scheduler with the learning rate initialized at 1e-4. The reasoning data follows a vanilla structured format of $\{question\}\backslash n\{cot\_answer\}$ with final answers encapsulated in $\backslash boxed\{\}$ notation. In scaling experiments, the mid-training token size is further expanded to 60B while maintaining the same data blend ratio.

Furthermore, we utilize identical SFT data across all models, thereby eliminating the introduction of additional variables and ensuring consistency in evaluation conditions. The SFT stage takes a batch size of 256 and an epoch size of 3, processing inputs up to 16,384 tokens to accommodate extended

---

**Algorithm 3** Weighted Dynamic Time Warping Distance

---

**Input:** Sequence $\mathbf{x} = (x_1, \ldots, x_n)$; Sequence $\mathbf{y} = (y_1, \ldots, y_m)$; Weight vector $\mathbf{w} = (w_1, \ldots, w_m)$; Distance function $\delta$
**Output:** Distance value

1: **if** $n = 0$ **or** $m = 0$ **then return** $1.0$
2: Initialize matrices $\mathbf{D}, \mathbf{W} \in \mathbb{R}^{(n+1) \times (m+1)}$ with zeros
3: **for** $i = 1$ **to** $n$ **do**:
4:     $D_{i,0} \leftarrow D_{i-1,0} + w_1 \cdot \delta(x_i, y_1); W_{i,0} \leftarrow W_{i-1,0} + w_1$
5: **for** $j = 1$ **to** $m$ **do**:
6:     $D_{0,j} \leftarrow D_{0,j-1} + w_j \cdot \delta(x_1, y_j); W_{0,j} \leftarrow W_{0,j-1} + w_j$
7: **for** $i = 1$ **to** $n$ **do**:
8:     **for** $j = 1$ **to** $m$ **do**:
9:         $d \leftarrow \delta(x_i, y_j)$
10:         **if** $D_{i-1,j-1} \leq D_{i,j-1}$ **and** $D_{i-1,j-1} \leq D_{i-1,j}$ **then**
11:           $(D_{\text{prev}}, W_{\text{prev}}) \leftarrow (D_{i-1,j-1}, W_{i-1,j-1})$
12:         **elif** $D_{i,j-1} \leq D_{i-1,j}$ **then**
13:           $(D_{\text{prev}}, W_{\text{prev}}) \leftarrow (D_{i,j-1}, W_{i,j-1})$
14:         **else** $(D_{\text{prev}}, W_{\text{prev}}) \leftarrow (D_{i-1,j}, W_{i-1,j})$
15:         $D_{i,j} \leftarrow D_{\text{prev}} + w_j \cdot d; W_{i,j} \leftarrow W_{\text{prev}} + w_j$
16: **return** $\frac{D_{n,m}}{W_{n,m}}$ **if** $W_{n,m} > 0$ **else** $0.0$

---

reasoning sequences. It is optimized by the Adam algorithm and initializes the learning rate at 5e-6 with cosine decay to zero.

The RL stage runs 300 steps for all models and applies the GSPO algorithm (Zheng et al., 2025). Its hyperparameters include a global batch size of 512, a rollout batch size of 256, and a single epoch, optimized by the Adam algorithm with a constant learning rate of 1.0e-6. The generation max sequence length is set to 32,768, with the sample number at 16, temperature and top_p both at 1, and GSPO-specific clip range between 0.0003 and 0.0004.

## C.2 EVALUATION

We conduct an evaluation of the SFT and RL performance on challenging mathematical reasoning tasks, similar to Balunović et al. (2025). For statistical robustness, we use 32 repetitions for AIME and HMMT, and 10 repetitions for BeyondAIME, once for MATH500.

## C.3 DATASETS

Table 5 shows the comparison with open-source CoT QA datasets.

Table 5: Comparison with open-source CoT QA datasets.

| Dataset | Target Domain | CoT | Date |
|---------|---------------|-----|------|
| JiuZhang3.0 | Mathematical Reasoning | Short-CoT | 2024 May. |
| OpenMathInstruct-2 | Mathematical Reasoning | Short-CoT | 2024 Oct. |
| MegaMathQA | Mathematical Reasoning | Short-CoT | 2025 Apr. |
| OpenR1-Math-220k | Mathematical Reasoning | Long-CoT | 2025 Feb. |
| AM-DeepSeek-R1-Distilled | General Reasoning | Long-CoT | 2025 Mar. |
| CoTP | Mathematical Reasoning | Long-CoT | 2025 |

# D RESULTS

## D.1 GENERAL PERFORMANCE

Table 6 shows the detailed comparison of general performance, corresponding to the general evaluation results in Table 1 in Section 3.2. Our CoTP achieves better general performance than KnowEdu and LongCoTPool, demonstrating that CoTP effectively expands the reasoning potential of foundation models in challenging mathematical reasoning tasks while maintaining general performance. Note that the performance gaps between CoTP and OpenR1-Math or AM-Distilled stem from the heterogeneous nature of LongCoTPool, which integrates OpenR1-Math, AM-Distilled, and BoostQA, with BoostQA exhibiting relatively weaker general performance. Since CoTP is selected from LongCoTPool, it inherits these performance gaps. However, this does not affect our objective and the main conclusions of our experiments.

Table 6: General performance comparison of models mid-trained on different datasets. The best and second-best are in bold and underlined, respectively. Abbreviations: W.G. = WinoGrande, H.S. = HellaSwag, BBH = Big-Bench.

| Dataset | MMLU | CMMLU | C-Eval | W.G. | H.S. | ARC-C | BBH | DROP | AVG. |
|---------|------|-------|--------|------|------|-------|-----|------|------|
| KnowEdu | 69.83 | 75.59 | 74.30 | 58.00 | 50.50 | 83.50 | 48.86 | 54.50 | 64.39 |
| BoostQA | 69.97 | 75.50 | 73.25 | 54.00 | 44.00 | 84.00 | 47.09 | 58.50 | 63.29 |
| JiuZhang3.0 | 68.71 | 74.94 | 74.51 | 61.50 | 50.50 | 85.00 | 45.37 | 54.00 | 64.32 |
| MegaMathQA | 68.80 | 75.43 | 73.71 | 62.00 | 47.50 | 85.00 | 45.85 | 60.00 | 64.79 |
| OMInstruct-2 | 68.76 | 75.39 | 74.29 | 62.00 | 50.00 | 85.50 | 49.45 | 64.50 | 66.24 |
| OpenR1-Math | 69.28 | 76.05 | 74.23 | 61.50 | 55.50 | 87.50 | 49.54 | 59.00 | 66.58 |
| AM-Distilled | 69.65 | 75.73 | 74.14 | 61.50 | 63.00 | 85.00 | 50.75 | 64.00 | 67.97 |
| LongCoTPool | 69.72 | 76.08 | 73.89 | 57.00 | 53.00 | 87.00 | 49.00 | 61.87 | 65.95 |
| CoTP (Ours) | 69.54 | 75.67 | 74.33 | 59.50 | 55.50 | 83.50 | 49.11 | 61.50 | 66.08 |

## D.2 PASS@K CURVES

As illustrated in Figure 7, the pass@k curves for both base and SFT models on the CoT dataset show a stable increase as $k$ becomes larger. In contrast, for the KnowEdu corpus, the pass@k values are notably lower without the application of SFT, but they improve once SFT is applied. This indicates that SFT ensures the model has a foundational ability to produce long CoT outputs, contributing to a more equitable evaluation across different datasets. Given the consistent relative trend observed before and after SFT, we exhibit our main results on the SFT models to ensure reliability and fairness in comparison.

## D.3 SCALING DETAILS

The specific accuracy values at 60B tokens in the scaling experiments are presented in Table 7.

Table 7: Accuracy of different models at 60B tokens in the experiments of data volume scale.

| Dataset | AIME 2025 | AIME 2024 | HMMT 2025 | BeyondAIME | MATH500 |
|---------|-----------|-----------|-----------|------------|---------|
| KnowEdu | 0.10±0.20 | 0.94±0.61 | 3.65±1.18 | 0.30±0.34 | 42.00±4.33 |
| LongCoTPool | 27.88±2.67 | 32.62±2.83 | 18.22±2.24 | 10.01±1.65 | 90.56±2.87 |
| CoTP (Ours) | 31.52±2.87 | 43.85±3.16 | 22.60±2.68 | 14.00±2.15 | 91.00±2.51 |

## D.4 TRUNCATION RATIO

The truncation ratios are shown in Table 8. We observe that the truncation rate is notably lower in long CoT datasets, and the truncation rate decreases further in RL compared to SFT.

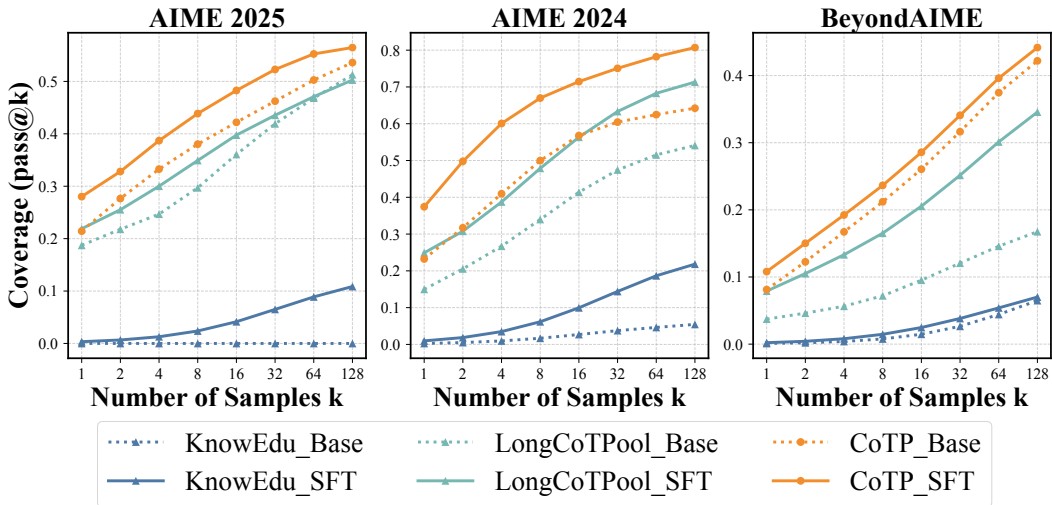

Figure 7: Pass@k curves of base and SFT models.

Table 8: Comparison of truncation ratio (%).

| Dataset | AIME 2025 | | AIME 2024 | | HMMT 2025 | | BeyondAIME | |
|---|---|---|---|---|---|---|---|---|
| | SFT | RL | SFT | RL | SFT | RL | SFT | RL |
| KnowEdu | 82.34 | 44.69 | 85.68 | 45.05 | 85.05 | 48.78 | 80.75 | 45.67 |
| LongCoTPool | 70.89 | 2.24 | 73.18 | 2.94 | 82.24 | 1.61 | 82.70 | 0.90 |
| CoTP (Ours) | 61.61 | 5.76 | 57.11 | 5.23 | 71.51 | 4.61 | 75.34 | 3.82 |

## D.5 RESPONSE LENGTH

As depicted in Figure 8, the response lengths during RL training exhibit different characteristics. Initially, models that have undergone SFT produce relatively longer responses. As the training progresses, the response lengths for LongCoTPool and CoTP stabilize and converge, whereas the lengths for KnowEdu continue to fluctuate significantly. This suggests that the RL training with KnowEdu may be prone to instability or collapse.

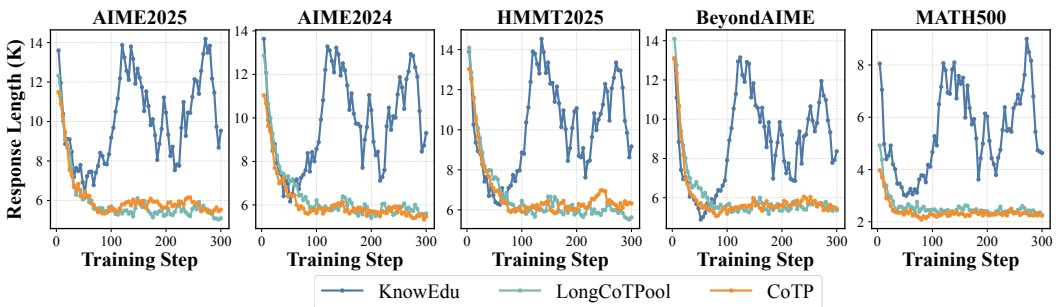

Figure 8: Response length (k) at the RL stage.

## D.6 EVALUATION CHART ON MORE BENCHMARKS.

Figure 9 and Figure 10 present the pass@k evaluation curves on the AIME2024 and Math500 benchmarks, respectively, illustrating the performance trajectory throughout the RL training pro-

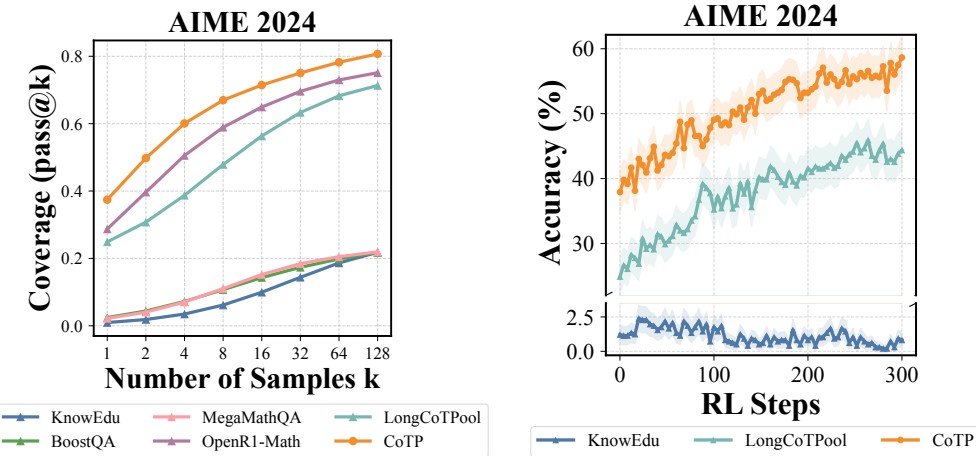

Figure 9: AIME2024.

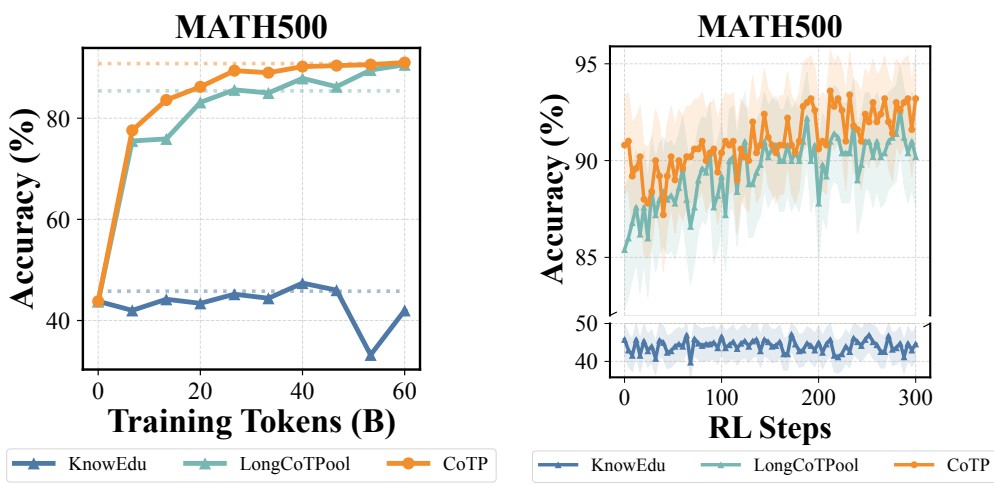

Figure 10: MATH500.

cess. These results serve as a complement to Figure 2 in Section 3.2 and are consistent with the results in Figure 2.

## E    REASONING PATTERN CHAIN ANNOTATION

**Prompt for CoT pattern chain extraction annotation**

Task Objective:
Systematically explore and summarize the Chain−of−Thought (CoT) processes employed by mainstream LLMs in reasoning tasks, analyzing the core reasoning patterns embedded within these processes.

Analysis Instructions:
Please conduct an in−depth examination of the reasoning paths taken by various AI models in reasoning tasks, demonstrating how different models approach and solve problems. Your goal is to summarize and categorize the general thinking patterns reflected in these

reasoning processes, to help understand the essential characteristics of CoT reasoning in large models.

Analysis Steps:
For each reasoning process, please clearly identify the following elements:
1. Use of keywords and high−frequency phrases
2. Logical structure and organization of argumentation
3. Techniques or strategies used to solve the problem
4. The manner in which reasoning steps are unfolded

Classification Requirements:
Based on the following commonalities, accurately categorize similar reasoning processes into one or more general reasoning patterns:
1. Lexical pattern (organization and use of common terms and phrases)
2. Logical framework (structure of argumentation and reasoning flow)
3. Solution pathway (methods and paths to reach conclusions)

Important Notes:
1. You are required to summarize "general thinking patterns for problem solving," not specific problem types.
2. Each pattern should be applicable to any problem scenario, not limited to a particular type of task.
3. Focus on the thinking method itself, rather than specific solution steps or answer content.

Illustrative Examples:
You may categorize as follows:
− Knowledge retrieval−based reasoning
− Reasoning combined with verification
− Step−by−step deductive calculation
− Detailed stepwise derivation
− etc.

Attention Points:
− Precisely categorize the above reasoning processes into one or more patterns (>=1), defining each category based on its shared characteristics, explaining its role in reasoning for the given case, and providing examples.
− Briefly explain your analysis and classification criteria first, then output detailed annotation for each reasoning pattern in the JSON format below. The "name" field for each pattern must be output in Chinese.
− The "pattern_chain" field outputs a list, where the element order represents the sequence of patterns used in this CoT solution, e.g., [1,2,3,4]. If necessary, the reasoning pattern chain may contain loops.
− Output atomic patterns only (no pattern should contain words like "and", "or", etc.).

Output Format:
```json
{
        "pattern_list": [
                {"id":1, "name": "", "description": "", "features": "", "
                sample_input_flow": "", "role_in_this_case": "", "
                corresponding_CoT_content": ["", ""], "common_elements": "", "
                typical_expressions": ["", ""] },
                {"id":2, "name": "", "description": "", "features": "", "
                sample_input_flow": "", "role_in_this_case": "", "
                corresponding_CoT_content": ["", ""], "common_elements": "", "
                typical_expressions": ["", ""] },
```

```
                ...
        ],
        "how_CoT_utilizes_patterns_in_this_case": {
                "process_description": "",
                "pattern_chain": [],
        },
}```

Reasoning process to be analyzed: {
        {extracted_responses}
}
```

# F  ANALYSIS

## F.1  REASONING PATTERNS ACROSS DIFFERENT MODELS

In the comparative analysis of reasoning patterns across different models on AIME 2025 & 2024, the number of patterns is defined as size of the entire set of reasoning patterns. Correct patterns refer to the set composed of patterns that appear in correct reasoning chains, while key patterns represent the overlap between the correct pattern sets and those mastered by DeepSeek-R1. Notably, DeepSeek-R1 has mastered 6,487 correct patterns, and the percentage indicates the proportion of the overlap within this set.

## F.2  PATTERN REPRESENTATION ANALYSIS

We conduct a comparative analysis of the pattern representations in Chinese and English. We sample pattern pairs from the CoTP dataset, manually selecting unrelated pairs that do not share core mathematical concept words. For example, "Problem Understanding and Information Extraction" versus "Trigonometric Identity Transformation" are considered unrelated because the former pertains to problem comprehension, while the latter involves a specific mathematical technique. In contrast, "Verification and Confirmation" and "Verification and Adjustment" are related, as both concern verification processes. We evaluate the similarity distributions of both related and unrelated pattern pairs in Chinese and English representations using the n-gram method.

As shown in Table 9 and Table 10, most pattern pairs in English are identified as either moderately similar or highly similar, making it difficult to differentiate between unrelated patterns, while Chinese offers a superior ability to distinguish between patterns. Compared with Chinese, English tends to display more misjudgment cases, as illustrated in Table 11.

These results demonstrate that Chinese pattern representation exhibits clear distinctions between unrelated pattern pairs under the character-level n-gram method, which is beneficial for the consistency and controllability of large-scale extraction of patterns. Even if there is slight noise or drift between patterns in Chinese, such as a small amount of character interference, this method is essentially insensitive to these variants and can still identify semantically similar patterns.

Table 9: The proportion of different similarity ranges of Chinese and English

| Similarity Range | Chinese | English |
|---|---|---|
| Low (<0.3) | 88.2% | 1.4% |
| Medium (0.3–0.7) | 11.2% | 83.2% |
| High (>0.7) | 0.6% | 15.4% |

Table 10: For the average similarity of unrelated pattern pairs, Chinese can score these pairs with low similarity, whereas English tends to misjudge them as similar.

| Language | Average Similarity |
|----------|--------------------|
| Chinese  | 0.035              |
| English  | 0.541              |

Table 11: Typical misjudgment cases with high similarity in English but low in Chinese.

| Chinese Pattern Pair | Chinese Similarity | English Pattern Pair | English Similarity | Root Cause |
|----------------------|--------------------|----------------------|--------------------|------------|
| 问题理解与信息提取 *vs* 三角恒等变换 | 0.000 | Problem Understanding and Information Extraction *vs* Trigonometric Identity Transformation | 0.762 | High repetition of suffixes "-tion", "-ation" |
| 数学建模与方程构建 *vs* 逻辑排除法 | 0.000 | Mathematical Modeling and Equation Construction *vs* Logical Elimination Method | 0.760 | Overlapping character-level suffixes |
| 验证与修正 *vs* 三角恒等变换 | 0.000 | Verification and Correction *vs* Trigonometric Identity Transformation | 0.758 | Common suffix interference |

### F.3 REASONING PATTERN VISUALIZATION IN DIFFERENT DOMAINS

To demonstrate the domain-agnostic nature of our CoTP framework, we visualize the reasoning patterns with high and low TF-IDF importance scores in Figure 11 across four representative STEM domains: physics, chemistry, biology, and computer science. These visualizations are generated from the TF-IDF weighted pattern distributions extracted from our training data.

These visualizations demonstrate that our pattern-based representation successfully captures domain-specific reasoning strategies across diverse STEM domains, supporting the claim that CoTP is applicable to any scenario that can be decomposed into atomic reasoning patterns.

## G CASE STUDY

In practice, we found that, unlike English, each character in Chinese possesses inherent semantic meaning, which makes Chinese particularly suitable for calculating distances between pattern entries using the ngram-cosine algorithm (see Algorithm 2). Consequently, we employ a annotation prompt (see Appendix E) to extract pattern chain features from the CoT data, and require the annotation model to output the patterns in Chinese, as described in Section 2.2.

For clarity in the presentation of experimental results, as illustrated in Figure 5, we have translated the pattern entries into English. The original Chinese-to-English mapping of pattern expressions is provided in Table 12.

Table 12: Mapping of English and Chinese expressions in reasoning models.

| Chinese Pattern | English Pattern | Chinese Pattern | English Pattern |
|-----------------|-----------------|-----------------|-----------------|
| 参数代入与计算 | Parameter Substitution | 数据格式转换 | Data Conversion |
| 分组因式分解 | Group Factorization | 几何图形优化 | Geometry Optimization |
| 中位数统计计算 | Median Calculation | 分段函数积分 | Piecewise Integration |
| 比例推理与分配 | Proportion Reasoning | 分数结构构造 | Fraction Construction |
| 线性组合构造 | Linear Combination | 数学项式计算 | Term Calculation |

Table 12: (continued)

| Chinese Pattern | English Pattern | Chinese Pattern | English Pattern |
|---|---|---|---|
| 阶段函数应用 | Step Function | 空间区域分割 | Space Partition |
| 公式直接代入 | Formula Substitution | 百分比数值转换 | Percentage Conversion |
| 分组变换方法 | Group Transform | 递归式展开 | Recursive Expansion |
| 逆向映射推理 | Inverse Mapping | 集合元素比较 | Set Comparison |
| 函数反演技巧 | Function Inversion | 代数项替换 | Term Replacement |
| 方向性映射 | Direction Mapping | 条件概率分析 | Conditional Probability |
| 数据排序整理 | Data Sorting | 指数模式匹配 | Exponential Matching |
| 货币单位转换 | Currency Conversion | 数量关系分析 | Relation Analysis |
| 凸包几何计算 | Convex Hull | 复合函数分解 | Function Decomposition |
| 不等式逐步解析 | Inequality Parsing | 路径方案枚举 | Path Enumeration |
| 模运算与替换 | Modular Substitution | 项数统计计数 | Term Counting |
| 基本原理应用 | Basic Principle | 欧拉线几何性质 | Euler Line Property |
| 线性变换操作 | Linear Transform | 公共因子提取 | Common Factor |
| 交叉相乘消元 | Cross Multiplication | 排除筛选法 | Exclusion Method |
| 集合穷举列举 | Set Enumeration | 多项式根系分析 | Polynomial Root Analysis |
| 题目提示利用 | Hint Utilization | 多解方案筛选 | Multi-solution Filter |
| 面积几何匹配 | Area Matching | 指数参数调整 | Exponential Adjustment |
| 算式化简优化 | Operation Simplification | 共轭根识别 | Conjugate Root ID |
| 逻辑推理验证 | Logic Reasoning | 问题分步分解 | Problem Decomposition |
| 数学工具运用 | Math Tools | 边界条件检验 | Boundary Check |
| 数值结果验证 | Numeric Verification | 结论确认总结 | Conclusion |
| 逻辑演绎推导 | Logic Deduction | 目标明确化 | Goal Clarification |
| 多方法交叉验证 | Multi-method Check | 数学概念应用 | Math Concepts |
| 结果整合输出 | Result Integration | 条件约束检查 | Condition Check |
| 问题抽象建模 | Problem Abstraction | 概念定义回顾 | Concept Definition |
| 知识检索调用 | Knowledge Retrieval | 规律泛化推广 | Generalization |
| 案例实例验证 | Case Verification | 反向逆推验证 | Reverse Verification |

## G.1 RELATIONSHIP BETWEEN REASONING PATTERNS AND REFLECTION

The cases that describe the relationship between the reflection and the chains of reasoning patterns are shown in Table 13, including the key features and descriptions of the reflection process in LLM reasoning.

## G.2 TOKEN ENTROPY

Figure 12 and Figure 13 present two examples of entropy-granular labeling of CoT data utilizing the open-source Qwen-7B model. High-entropy tokens frequently reflect the model's underlying reasoning process. Our framework exploits the fine-grained token-level entropy distribution to identify nuanced reasoning patterns within the model's CoT data.

## G.3 PROBLEM, REASONING CHAIN, AND EXTRACTED PATTERNS

Table 14: Examples of questions, CoT sequences, pattern chains, and entropy chains.

| Case 1 |
| --- |
| *Question* |

Table 14: (continued)

---

Imagine as 2015 fractions:

$$\frac{2}{2016}, \frac{3}{2015}, \frac{4}{2014}, \ldots, \frac{2014}{4}, \frac{2015}{3}, \frac{2016}{2}$$

Is it possible to choose three of these fractions with a product equal to 1?

*Pattern Chain*

Pattern Recognition and Structural Analysis $\rightarrow$ Equation Establishment and Transformation $\rightarrow$ Symmetry Analysis and Special Value Verification $\rightarrow$ Proof by Contradiction and No-Solution Exploration

*Chain-of-Thought*

Okay, let's see. The problem is asking if it's possible to choose three fractions from the given 2015 fractions such that their product equals 1. The fractions are structured in a specific way: starting from 2/2016, then 3/2015, 4/2014, and so on, up to 2016/2. Each numerator increases by 1 while the denominator decreases by 1.

First, I need to understand the pattern here. The first fraction is 2/2016, the next is 3/2015, then 4/2014, continuing until 2016/2. So, the nth fraction would be (n+1)/(2017 - n), right? Let me check. For n=1, (1+1)/(2017 - 1) = 2/2016, which matches. For n=2, 3/2015, which also matches. So yes, each fraction can be represented as (k+1)/(2017 - k) where k ranges from 1 to 2015. Alternatively, maybe another way to index them? Let me see. If we let the numerator be m, then m starts at 2 and goes up to 2016, and the denominator is 2016 - (m - 2) = 2018 - m. Wait, because when numerator is 2, denominator is 2016; when numerator is 3, denominator is 2015, which is 2016 -1, so denominator for numerator m is 2018 - m. Because 2018 - 2 = 2016, 2018 - 3 = 2015, etc. So each fraction is m/(2018 - m), where m ranges from 2 to 2016. That seems correct.

So the fractions can be written as m/(2018 - m) for m = 2, 3, ..., 2016. So there are 2015 fractions total because 2016 - 2 + 1 = 2015.

We need to find three fractions such that their product is 1. Let's denote the three fractions as m1/(2018 - m1), m2/(2018 - m2), m3/(2018 - m3). Then their product is: (m1 × m2 × m3) / [(2018 - m1)(2018 - m2)(2018 - m3)] = 1

Therefore, the equation simplifies to: m1 × m2 × m3 = (2018 - m1)(2018 - m2)(2018 - m3)

So, we need three numbers m1, m2, m3 (each between 2 and 2016 inclusive) such that the product of the three m's equals the product of (2018 - m) for each m.

This seems like a symmetrical equation. Maybe we can look for some symmetry here. Let's consider that for each m, (2018 - m) is also a term in the fractions. For example, if m is 2, then 2018 - m = 2016, which is the denominator of the first fraction. Similarly, if m is 2016, then 2018 - 2016 = 2, which is the denominator of the last fraction. So each fraction m/(2018 - m) has a reciprocal fraction (2018 - m)/m. However, note that when m is 2018 - m, which would be when m = 1009, then the fraction is 1009/1009 = 1. But looking at the original fractions, starting from 2/2016 up to 2016/2, the middle fraction would be when m is around 1009. Let me check: since there are 2015 fractions, the middle one is at position 1008, but maybe m is 1009. Let's compute 2018 - m = 2018 - 1009 = 1009, so yes, when m=1009, the fraction is 1009/1009=1. So that fraction is 1, so multiplying by that fraction doesn't change the product. But the problem is asking for three fractions, so even if one of them is 1, the other two need to multiply to 1 as well. But 1 is only present once (if m=1009 is allowed). Wait, m ranges from 2 to 2016, so m=1009 is included. So the fraction 1009/1009 is part of the list. So if we include that, then maybe we can have two fractions that are reciprocals. For example, take 2/2016 and 2016/2. Their product is (2×2016)/(2016×2) = 1. Then if we include 1, the product would be 1×1=1. Wait, but we need three fractions. So 2/2016 × 2016/2 × 1009/1009 = 1. So that's three fractions. But hold on, 2/2016, 2016/2, and 1009/1009. Are these all in the list? Yes. The first fraction is 2/2016, the last is 2016/2, and the middle one is 1009/1009. So their product is 1. So that would be a valid triplet. But wait, but the problem says "three of these fractions". So if 1009/1009 is allowed, then yes. But is 1009/1009 in the list? Let's check. When m=1009, denominator is 2018 - 1009 = 1009. So yes, that's the fraction 1009/1009. So that's in the list. Therefore, the triplet is 2/2016, 2016/2, and 1009/1009. So their product is (2 × 2016 × 1009)/(2016 × 2 × 1009) = 1. So yes, that works.

---

Table 14: (continued)

Wait, but the problem is stated as: Is it possible to choose three of these fractions with a product equal to 1? **...**

***Token Entropy***
{Okay: 1.0781}, {,: 0.0693}, { let: 1.3125}, {'s: 0.1729}, { see: 2.7188}, {.: 0.7969}, { The: 1.8203}, { problem: 2.0938}, { is: 1.9375}, { asking: 1.0859}, { if: 1.0938}, { it: 1.2656}, {'s: 0.5078}, { possible: 0.0173}, { to: 0.0376}, { choose: 1.1641}, { three: 0.0491}, { fractions: 0.8320}, { from: 0.2393}, { the: 0.1699}, { given: 1.2109}, { : 1.0313}, {2: 0.3691}, {0: 0.0021}, {1: 0.0044}, {5: 0.3613}, { fractions: 0.3047}, { such: 1.4297}, { that: 0.0027}, { their: 0.1670}...

**Case 2**

***Question***
Let $a, b, c$ be positive integers, and the quadratic equation $ax^2 + bx + c = 0$ has two real roots whose absolute values are both less than $\frac{1}{3}$. Find the minimum value of $a + b + c$.

***Pattern Chain***
Condition Analysis and Constraint Extraction $\rightarrow$ Mathematical Derivation and Inequality Analysis $\rightarrow$ Systematic Enumeration and Verification $\rightarrow$ Two-Root Case Analysis

***Chain-of-Thought***
Okay, so I need to find the minimum value of a + b + c where a, b, c are positive integers, and the quadratic equation $ax^2 + bx + c = 0$ has two real roots whose absolute values are both less than 1/3. Hmm, let's start by recalling some properties of quadratic equations and their roots.

First, for a quadratic equation to have real roots, the discriminant must be non-negative. The discriminant D is $b^2 - 4ac$. So, $D \geq 0$. That gives us one condition: $b^2 \geq 4ac$.

Next, the roots have absolute values less than 1/3. Let me denote the roots as $r_1$ and $r_2$. Then $|r_1| < 1/3$ and $|r_2| < 1/3$. Since the quadratic is $ax^2 + bx + c$, by Vieta's formulas, we have:
$r_1 + r_2 = -b/a$ and $r_1 \times r_2 = c/a$

Given that the absolute values of the roots are less than 1/3, we can derive some inequalities. Let's think about the implications. Since both roots are real and their absolute values are less than 1/3, they must lie in the interval (-1/3, 1/3). So both roots are between -1/3 and 1/3.

But since coefficients a, b, c are positive integers, let's consider the signs of the roots. If the quadratic equation has two real roots, and a, b, c are positive, then the quadratic opens upwards (since a > 0). The sum of the roots is -b/a, which is negative because b and a are positive. The product of the roots is c/a, which is positive because c and a are positive. Therefore, both roots must be negative. So, actually, the roots are in the interval (-1/3, 0). That simplifies things a bit because we don't have to consider positive roots. Both roots are negative, so their absolute values are just their magnitudes.

Therefore, we can say that the roots $r_1$ and $r_2$ satisfy $-1/3 < r_1 \leq r_2 < 0$. Let me rephrase Vieta's formulas in terms of magnitudes. Let me set $r_1 = -s$ and $r_2 = -t$ where s and t are positive numbers less than 1/3. Then:
$r_1 + r_2 = -(s + t) = -b/a \Rightarrow s + t = b/a$ and $r_1 \times r_2 = st = c/a$

So s and t are both in (0, 1/3). So we have $s + t = b/a$ and $st = c/a$. Then, since a, b, c are positive integers, we need to find a, b, c such that these equations hold with s and t ¡ 1/3.

Our goal is to minimize a + b + c. Since a, b, c are positive integers, the minimal value is likely to be small, so maybe a, b, c are small integers. But let's think through systematically.

Given that s and t are both less than 1/3, so $s + t < 2/3$ and $st < (1/3)(1/3) = 1/9$. But $s + t = b/a$ and $st = c/a$.

So from $st = c/a$, since $st < 1/9$, then $c/a < 1/9 \Rightarrow c < a/9$. But since c is a positive integer, this gives that a must be at least 10 for c to be at least 1 (since $1 < a/9 \Rightarrow a > 9$). Wait, but is that correct? Let's see: If $c/a < 1/9$, then $c < a/9$. Since c must be at least 1, then $1 \leq c < a/9 \Rightarrow a > 9c$. But since c is at least 1, a must be greater than 9. So the minimal possible a is 10? Wait, but maybe c can be 1 and a = 10, but then c = 1, a = 10. Then $1 < 10/9 \approx 1.11$, so that would satisfy $1 < 10/9$. But 1 is not less than 10/9, it's equal? Wait, no. Wait, c is less than a/9. So $1 < a/9 \Rightarrow a > 9$. So $1 < a/9$.**...**

***Token Entropy***

Table 14: (continued)

---

{Okay: 0.8789}, {,: 0.1445}, { so: 1.6484}, { I: 1.4453}, { need: 1.9688}, { to: 0.0149}, { find: 0.8711}, { the: 0.3203}, { minimum: 0.3379}, { value: 0.1523}, { of: 0.0513}, { a: 0.9844}, { +: 1.9609}, { b: 0.0005}, { +: 0.0004}, { c: 0.0007}, { where: 1.4141}, { a: 0.9961}, {,: 0.0122}, { b: 0.0011}, {,: 0.0127}, { c: 0.5039}, { are: 0.0103}, { positive: 0.1445}, { integers: 0.0110}, {,: 0.7500}, { and: 0.1797}, { the: 0.2676}, { quadratic: 0.3613}, { equation: 0.2051}**...**

---

**Case 3**

---

*Question*
Symmetric about the $y$-axis, passing through the focus of the parabola $y^2 = 4x$, and divided into two arcs with a length ratio of $1:2$ by the line $y = x$, the equation of the circle is

---

*Pattern Chain*
Knowledge Retrieval and Verification $\rightarrow$ Symmetry Analysis $\rightarrow$ Equation Construction and Solution $\rightarrow$ Geometric Condition Transformation $\rightarrow$ Vector Analysis $\rightarrow$ Multiple Solution Verification

---

*Chain-of-Thought*
Okay, let me try to work through this problem step by step. So, the problem is about finding the equation of a circle that's symmetric about the y-axis, passes through the focus of the parabola $y^2 = 4x$, and is divided into two arcs by the line $y = x$ with a length ratio of 1:2. Hmm, that seems a bit involved, but let's break it down.

First, let's recall some basics. The parabola $y^2 = 4x$. The standard form of a parabola that opens to the right is $y^2 = 4ax$, where $(a, 0)$ is the focus. Comparing that to the given equation, $4a = 4$, so $a = 1$. Therefore, the focus of this parabola is at $(1, 0)$. So the circle we're looking for passes through $(1, 0)$. Got that.

Next, the circle is symmetric about the y-axis. That means if $(x, y)$ is on the circle, then $(-x, y)$ is also on the circle. So the equation of the circle must be of the form $(x - h)^2 + (y - k)^2 = r^2$, but since it's symmetric about the y-axis, $h$ must be 0. Because if the center is $(h, k)$, then symmetry about the y-axis implies that $h = -h$, so $h = 0$. Therefore, the equation simplifies to $x^2 + (y - k)^2 = r^2$. So the center is at $(0, k)$ and radius $r$. So we need to find $k$ and $r$.

We know the circle passes through the focus $(1, 0)$. Let's plug that point into the equation to get an equation relating $k$ and $r$. Plugging in $x = 1$, $y = 0$:
$1^2 + (0 - k)^2 = r^2 \Rightarrow 1 + k^2 = r^2$. That's one equation.

Now, the other condition is that the line $y = x$ divides the circle into two arcs with a length ratio of 1:2. Hmm, okay. So the circumference of the circle is $2\pi r$, and the line $y = x$ intersects the circle at two points, dividing it into two arcs. The ratio of their lengths is 1:2, so the shorter arc is $(1/3) \times 2\pi r$ and the longer arc is $(2/3) \times 2\pi r$. So the central angles corresponding to these arcs should be 120 degrees and 240 degrees, since the ratio 1:2 corresponds to angles of 120 and 240 (which sum to 360). So the angle between the two intersection points as viewed from the center of the circle should be 120 degrees.

Alternatively, maybe the line $y = x$ cuts the circle such that the arcs are in a 1:2 ratio. To find the central angle corresponding to each arc, since arc length is proportional to the central angle. So if the ratio is 1:2, the angles are $\theta$ and $2\theta$, where $\theta + 2\theta = 360$, so $\theta = 120$. Wait, no. Wait, 1:2 ratio in terms of length. So total circumference is 3 parts, so each part is 120, 240. So the central angles are 120 and 240, so the central angle between the two intersection points is 120, right?**...**

---

*Token Entropy*
{Okay: 1.1641}, {,: 0.1143}, { let: 1.1484}, { me: 0.1060}, { try: 2.5000}, { to: 0.3301}, { work: 1.7969}, { through: 1.0313}, { this: 0.5078}, { problem: 0.9023}, { step: 0.4648}, { by: 0.6523}, { step: 0.0007}, {.: 1.3359}, { So: 1.8438}, {,: 1.0781}, { the: 1.3672}, { problem: 2.5938}, { is: 2.5938}, { about: 1.5391}, { finding: 0.7813}, { the: 0.4668}, { equation: 0.0344}, { of: 0.0060}, { a: 0.0527}, { circle: 0.0757}, { that: 0.8398}, {'s: 1.2188}, { symmetric: 0.2373}, { about: 0.1069}**...**

(a) Physics: high importance scores.

(b) Physics: low importance scores.

(c) Chemistry: high importance scores.

(d) Chemistry: low importance scores.

(e) Biology: high importance scores.

(f) Biology: low importance scores.

(g) Computer science: high importance scores.

(h) Computer science: low importance scores.

Figure 11: Examples of reasoning patterns across four STEM domains. For each domain, the left column (red) shows patterns with high importance scores, while the right column (blue) displays patterns with low importance scores.

Table 13: The relationship between the reflection and the chain of reasoning patterns.

| Key Features | Description | Chains of Reasoning Patterns |
|---|---|---|
| Self-supervision and Self-correction | The model can actively review its own output, discover and correct logical, factual or expressive errors. This self-monitoring ability is at the core of the reflection process. | Structural Analysis and Counting, **Cross-validation and Error Correction**, Reaction Mechanism Deduction, Conditional Driven Path Selection, Structural Analysis and Counting, Reaction Mechanism Deduction, **Cross-validation and Error Correction** |
| Multi-step Iteration | Reflection is not a one-off event but can be carried out in multiple rounds. The model can repeatedly check and correct its reasoning process until it outputs a more reasonable and accurate result. | Formula Substitution Method, Equation Simplification Method, **Iterative Trial and Error Method**, Verification Method |
| Explicit Reasoning Chain | During the reflection process, models often explicitly review and analyze their reasoning steps (such as "chain of thought"), looking for the rationality and potential problems of each step. | **Pattern Recognition and Analogical Reasoning**, Construction of Recursive Relations, Exact Computation and Verification, Approximate Estimation and Boundary Determination, **Error Checking and Correction** |
| Dynamic Adjustment | The model will dynamically adjust the reasoning direction, supplement information or change the reasoning strategy based on the reflection results to adapt to different problems and scenarios. | Variable Substitution and Recombination, Step-by-step Decomposition and Simplification, Substitution of Known Results, Numerical Verification and Confirmation, **Multi-method Cross-verification** |
| Uncertainty Identification | During the reflection process, the model can identify uncertainties or knowledge blind spots in its own reasoning and attempt to make up for these deficiencies. | Knowledge Retrieval Reasoning, **Uncertainty Processing Reasoning**, Stepwise Computation Reasoning, Conditional Verification Reasoning |
| Enhanced Interpretability | Through reflection, the model can output a more detailed reasoning process and correction reasons, enhancing the interpretability and transparency of the answers. | Understanding of Physical Processes, Derivation of Threshold Conditions, Verification of Mathematical Calculations, **Interpretation and Application of Results** |
| Recursive Reasoning | Recursive reasoning means that after the model generates the output for the first time, it will return to certain key steps, concepts or assumptions to conduct reasoning again to verify or adjust the previous inference. This reflective process usually occurs in more complex tasks, especially in scenarios that require multiple rounds of reasoning. | Problem Decomposition and Sub-goal Setting, **Mathematical Structure Abstraction**, **Logical Equivalence Transformation**, Special Case Verification and Induction, Combinatorial Counting Technique, **Mathematical Structure Abstraction**, **Logical Equivalence Transformation**, Combinatorial Counting Technique |

Okay, let's see here. We have six points A, B, C, D, E, F on a straight line in that order. Point G is not on the line. We need to find the area of triangle BGE. First, maybe I should sketch the points on a line to visualize their positions. Let me note down the given distances: AC = 26, BD = 22, CE = 31, DF = 33, AF = 73. Also, CG = 40 and DG = 30. Hmm. The coordinates of G are unknown, but since it's not on the line, we might need to assign coordinates to the points A through F first, then figure out G's coordinates using the distances CG and DG. Then, once we have coordinates for B, G, and E, we can compute the area of triangle BGE. Right, coordinate geometry approach makes sense here. Let's assign coordinates. Since all points are collinear, let's place them on the x-axis for simplicity. Let me let point A be at coordinate 0. Then, since AF = 73, point F would be at coordinate 73. Wait, but if A is at 0 and F is at 73, the other points B, C, D, E are in between. But need to figure out their positions based on given distances AC, BD, CE, DF. Let me denote the coordinates of A, B, C, D, E, F as a, b, c, d, e, f respectively. Since they are in order A, B, C, D, E, F on a straight line, their coordinates should satisfy a < b < c < d < e < f. And we know a = 0, f = 73. Given AC = 26, so c - a = 26 → c = 26, since a = 0. Similarly, BD = d - b = 22, CE = e - c = 31, DF = f - d = 33. But wait, DF is given as 33. Since f = 73, then DF = 73 - d = 33 → d = 73 - 33 = 40. So d = 40. Then, BD = d - b = 22 → 40 - b = 22 → b = 40 - 22 = 18. Therefore, b = 18. Next, CE = e - c = 31. c was found as 26, so e = 26 + 31 = 57. Therefore, e = 57. Let's verify the positions: a = 0, b = 18, c = 26, d = 40, e = 57, f = 73. Let's check the other distances if they make sense. AF = f - a = 73 - 0 = 73, correct. BD = 40 - 18 = 22, correct. CE = 57 - 26 = 31, correct. DF = 73 - 40 = 33, correct. So, the coordinates along the x-axis are: A: 0, B:18, C:26, D:40, E:57, F:73. Now, point G is not on this line. We know CG = 40 and DG = 30. So, points C and D are on the x-axis at 26 and 40 respectively. G is somewhere off the x-axis such that the distance from C(26,0) to G is 40, and the distance from D(40,0) to G is 30. Therefore, we can consider G as a point with coordinates $(x,y)$, $y \neq 0$, satisfying: $\sqrt{(x - 26)^2 + y^2} = 40$ --> $(x - 26)^2 + y^2 = 1600$ ...(1) and $\sqrt{(x - 40)^2 + y^2} = 30$ --> $(x - 40)^2 + y^2 = 900$ ...(2) We can subtract equation (2) from equation (1): $[(x - 26)^2 + y^2] - [(x - 40)^2 + y^2] = 1600 - 900$ $(x - 26)^2 - (x - 40)^2 = 700$ Expanding both squares: $(x^2 - 52x + 676) - (x^2 - 80x + 1600) = 700$ Simplify: $x^2 - 52x + 676 - x^2 + 80x - 1600 = 700$ Combine like terms: $(-52x + 80x) + (676 - 1600) = 700$ $28x - 924 = 700$ $28x = 700 + 924 = 1624$ $x = 1624 / 28 =$ Divide numerator and denominator by 4: $406 / 7 → 58.0$? Wait, wait. Let me compute 28 × 58 = 1624. Yes, 28 × 58: 28×50=1400, 28×8=224, total 1624. Therefore, x=58. Wait, but points D is at 40 on the x-axis, and E is at 57, F is at 73. So x=58 is between E and F? Hmm, but G could be there, off the line. But let's check. Wait, coordinates are x=58, so E is at 57, so G is at x=58 on the x-axis, but since y ≠0, but the x-coordinate is 58. So, the distance from C(26,0) to (58,y): sqrt( (58-26)^2 + y^2 ) = sqrt(32² + y² ) = sqrt(1024 + y² ) which is given as 40. So, 1024 + y² =1600 → y² = 576 → y=24 or y=-24. Similarly, distance from D(40,0) to (58,y): sqrt( (58-40)² + y² ) = sqrt(18² + y² )= sqrt(324 + y² )=30. Therefore, 324 + y²=900 → y²=576, same as before. So y=24 or -24. Therefore, G is at (58,24) or (58,-24). Since area is absolute, it won't matter if y is positive or negative. So take y=24. So coordinates of G are (58,24). Now, points B, G, E. Coordinates: B is at (18,0), G at (58,24), E at (57,0). Wait, E is at 57 on the x-axis. So E is (57,0). B is (18,0). G is (58,24). To find the area of triangle BGE. Coordinates: B: (18, 0) G: (58, 24) E: (57, 0) We can use the formula for the area of a triangle given three coordinates. The formula is $|(x1(y2 - y3) + x2(y3 - y1) + x3(y1 - y2))/2|$. Plugging in: x1 = 18, y1 = 0 x2 = 58, y2 =24 x3 =57, y3=0 Compute each term: 18*(24 - 0) = 18*24 = 432 58*(0 - 0) =58*0=0 57*(0 -24)=57*(-24)= -1368 Sum: 432 +0 -1368 = -936 Take absolute value and divide by 2: | -936 | /2=936/2=468 Therefore, area is 468. Wait, is this correct? Let me check. Alternatively, since points B and E are on the x-axis, and G is above, the base can be BE and the height is the vertical distance from G to BE. Let me check BE length. BE is from 18 to57 on x-axis, so length 57 -18=39. The height is the y-coordinate of G, which is24. Therefore, area is (1/2)*base*height= (1/2)*39*24. Calculate: 39*24= (40-1)*24=40*24 -24=960-24=936. Then 936/2=468. Yes, same result. So the area is 468. So maybe \boxed{468} is the answer. But wait, the problem states "Find the area of triangle BGE". Let me double-check. Coordinates: B (18,0), G (58,24), E (57,0). If I use the shoelace formula: List the coordinates in order: (18,0), (58,24), (57,0), (18,0). Compute sum of products down diagonals: 18*24 +58*0 +57*0 = 432 +0 +0 = 432 Sum of products up diagonals: 0*58 +24*57 +0*18 = 0 + 1368 +0=1368 Subtract the two: |432 -1368|=936 Area = 936/2=468. So same answer. So that's confirmed. Therefore, the area is 468. **Final Answer** \boxed{468} </think> We have six points \( A, B, C, D, E, \) and \( F \) lying on a straight line in that order, and a point \( G \) not on the line. We need to find the area of triangle \( \triangle BGE \). First, we assign coordinates to the points \( A, B, C, D, E, \) and \( F \) on the x-axis. Let \( A \) be

...

| Low (0.00) | High (4.72) |

Figure 12: Example 1 of token entropy chain annotation for CoT data High-entropy tokens frequently indicate the underlying reasoning process of the model.

Okay, so I have this problem here: There exist real numbers x and y, both greater than 1, such that log base x of y^x equals log base y of x^{4y} equals 10. I need to find the product xy. Hmm, let's start by breaking down the given equations. First, let's recall the logarithm properties. The expression log base a of b^c can be rewritten using the power rule: log_a(b^c) = c * log_a(b). So applying that here might simplify things. Starting with the first equation: log_x(y^x) = 10. Using the power rule, this becomes x * log_x(y) = 10. Similarly, for the second equation: log_y(x^{4y}) = 10. Applying the power rule here gives 4y * log_y(x) = 10. Alright, so now we have two equations: 1) x * log_x (y) = 10  2) 4y * log_y(x) = 10  I know that log_a(b) is the reciprocal of log_b(a). That 's another logarithm property: log_a(b) = 1 / log_b(a). So, if I denote log_x(y) as A, then log_y(x) = 1/A. Let's assign variables to make this easier. Let me set A = log_x(y ), so the first equation becomes x * A = 10. Then the second equation becomes 4y * (1/A) = 10, because log_y(x) = 1/A. So, equations now:  1) x * A = 10  2) (4y)/A = 10  Now, maybe solve for one variable in terms of another. Let's solve the first equation for x: x = 10 / A. Similarly, from the second equation: (4y)/A = 10 => 4y = 10A => y = (10A)/4 => y = ( 5A)/2.  So, now we have expressions for x and y in terms of A:  x = 10 / A  y = (5A)/2  Our ultimate goal is to find xy. Let's compute xy with these expressions:  xy = (10 / A) * (5A / 2) = (10 * 5A) / (2A) = (50A) / (2A) = 25. Wait, the A terms cancel out here. So, xy = 25?  But that seems too straightforward. Wait, but if the problem states that both log expressions equal 10 and by solving these equations we get xy = 25. Maybe this is the case. Let me check again.  But hold on, maybe there's something wrong here. Let's verify if x and y as functions of A satisfy the original equations. Let's see, x = 10/A, y = 5A/ 2. Let's compute log_x(y^x) and see if it's 10.  First, log_x(y^x). Using the power rule: x * log_x(y). But log_x(y) is A, so x * A. x * A = (10/A) * A = 10. That checks out.  Now the other equation: log_y(x^{4y}) = 4y * log_y(x). Log_y(x) is 1/A, so 4y*(1/A). Substit uting y = 5A/2: 4*(5A/2)*(1/A) = 4*(5/2) = 10. That also checks out.  So even though A is involved in the expressions, when you plug everything back in, the A's cancel out, and both conditions give 10. Therefore, regardless of A (as long as A is positive, since x and y are greater than 1), the product xy is 25. Wait, but A is log_x(y). Since x and y are greater than 1, what constraints does that put on A?  Well, if x > 1 and y > 1, then log_x(y) can be greater than 1 or less than 1 depending on whether y is greater than x or not. For example, if y = x^k, then log_x(y) = k. So, if k > 1, then y > x, if k =1, y = x , if k <1, then y < x.  But here, since x and y are both greater than 1, but we can have different relations between x and y. But in our solution, xy=25, regardless of A. But wait, does this mean that for any A (positive, as log is defined), x and y would adjust to satisfy the equations, and their product is always 25?  But we need to ensure that x > 1 and y>1. So let's check what conditions on A will make x and y greater than 1.  From x = 10/A >1, so 10/A >1 => A <10. Similarly, from y = 5A/2 >1, so 5A/2 >1 => A > 2/5.  So, A must be between 2/5 and 10 for x and y to be greater than 1. So even though A can vary in that interval, the product xy is fixed at 25. So the answer is 25. Wait, but let me think again. Since both the logs result in 10, which is a fixed value, even though variables are involved, the product is determined uniquely.  Alternatively, perhaps there's a different way to approach the problem. Let me try changing the base of the logarithms to see if we can relate x and y more directly.  Let's first note the two original equations :  1) log_x(y^x) = 10. As before, this is x * log_x(y) = 10.  2) log_y(x^{4y}) = 10. This is 4y * log_y(x) = 10.  Let's switch the base of logarithm to natural log to see if that helps. Recall that log_a(b) = ln(b)/ln(a).  So, log_x(y) = ln(y)/ln(x), and log_y(x) = ln (x)/ln(y).  So substituting into equation 1:  x * (ln y / ln x) = 10  Equation 2:  4y * ( ln x / ln y) = 10  Let's denote u = ln x and v = ln y.  Then equation 1 becomes:  x * (v / u) = 10  Equation 2 becomes:  4y * (u / v) =10  But we can express x and y in terms of exponentials:  x = e^{u}, y = e^{v}  So substituting back into equation 1:  e^{u} * (v / u) = 10 --> e^u * v =10 u  Equation 2:  4 e^{v} * (u / v) =10 --> e^v * u = (10 /4 ) v = 2.5 v  So now we have two equations:  1) e^u * v =10 u  2) e^v * u =2.5 v  This seems a bit complicated. But maybe we can relate them. Let's denote equations as:  From 1) v = (1 0 u) / e^u  Plug this into equation 2: e^v * u =2.5 * (10 u / e^u ) => e^v * u =25 u / e^u  Cancel u (assuming u ≠0, which it isn't since x >1, so ln x = u >0)  Therefore, e^v =25 / e^u => e^{u + v}=25 => u + v = ln25  So, u + v = ln25. But since u = ln x and v= ln y, then ln x + ln y= ln25 => ln(xy)=ln25 => xy=25.  So, that gives us xy=25, regardless of the values of u and v. So that's a different way to look at it, and it shows that regardless of the specific x and y, their product must be 25.  Therefore, the answer is 2 5.  Wait, this is the same answer as before, but now through this method, which is more direct because when we add the exponents, it relates to the product of x and y. So that method seems to get directly to the answer.  Alternatively, when I used substitution with variable A, leading to xy=25, but also checked constraints on A to ensure that x and y are greater than 1, but regardless of those constraints, as long as they are met, the

···

Low (0.00)                                                                                    High (4.47)

Figure 13: Example 2 of token entropy chain labeling. High-entropy tokens frequently indicate the underlying reasoning process of the model.

