# OpenReview forum: "Expanding Reasoning Potential in Foundation Model by Learning Diverse Chains of Thought Patterns"
_ICLR.cc/2026/Conference — ICLR 2026 Poster_

### Official Review · Reviewer_5gwE · 2025-10-28

**Soundness:** 3
**Presentation:** 3
**Contribution:** 3
**Rating:** 6
**Confidence:** 3

**Summary:**

The paper introduces CoTP, a framework for expanding the reasoning potential of foundation models by selecting high value chain of thought (CoT) data. The authors define “reasoning potential” as the inverse of expected attempts to solve a problem, extract atomic reasoning patterns to build a core set, and use a dual granularity similarity measure (pattern chains + token entropy) with a Hungarian assignment algorithm to select training data. Experiments on an 85A6B MoE model show significant gains on challenging math benchmarks and improved RL upper bounds.

**Strengths:**

1. Provides a formal definition of reasoning potential, linking it to expected attempts and offering theoretical grounding.

2. Proposes a novel data selection method combining reasoning pattern chains and token entropy, operationalized with weighted DTW and Hungarian matching.

3. Demonstrates strong empirical improvements on difficult mathematical reasoning tasks while maintaining general performance.

**Weaknesses:**

1. The construction of the core set heavily depends on strong model annotations and manual filtering, which may introduce bias, limit scalability, and lead to redundancy.

2. The similarity computation (weighted DTW combined with the Hungarian algorithm) could be computationally expensive when applied to very large data pools, raising concerns about efficiency.

3. The reliance on Chinese‑expressed reasoning patterns for distance computation lacks empirical justification, and the paper does not clearly explain how consistency and quality of the large set of abstracted patterns are maintained at scale.

**Questions:**

1. Could the authors clarify how they mitigate potential bias and scalability issues in the core set construction process, and provide an analysis of the computational cost of their similarity matching pipeline at scale?

2. The paper relies on reasoning patterns expressed in Chinese to compute distances between pattern chains. However, it is unclear whether there is empirical evidence demonstrating that this language‑specific representation is optimal or robust. Could the authors provide experiments or ablations to justify this choice?

3. Moreover, when abstracting a large number of reasoning patterns from massive datasets, how do the authors ensure that the extracted patterns remain consistent, controlled, and aligned with the intended definition of “high‑value” reasoning steps? What mechanisms are in place to prevent noise or drift in the pattern inventory?

4. The proposed CoTP framework shows consistent and significant improvements on mathematical reasoning benchmarks, yet it does not yield comparable gains on general‑domain tasks, sometimes even underperforming other baselines. Could the authors clarify why the method appears highly effective in math reasoning but does not extend to broader general benchmarks? Are there inherent limitations in the design of CoTP that make it domain‑specific?

---

> ### Author Response · Authors · 2025-11-21
> **Response to Reviewer 5gwE (part 1/3)**
>
> Thank you for your feedback and support of our work. Please find our responses to your questions below.
>
> >### `W1&Q1: The construction of the core set heavily depends on strong model annotations and manual filtering, which may introduce bias, limit scalability, and lead to redundancy. / Could the authors clarify how they mitigate potential bias and scalability issues in the core set construction process?`
>
> **Our core set construction is designed with scalability.** Specifically:
> 1. **LLMs provide reliable and efficient annotation.** Current state-of-the-art models demonstrate over 97% agreement with professional annotators in labeling tasks, and they are significantly more cost-effective than manual annotation, becoming a mainstream approach [1].
> 2. **Manual screening was conducted with the assistance of statistical non-manual methods to ensure core set quality while minimizing subjectivity.** During question screening, manual intervention was limited to quality and answer correctness verification. Specifically, we employed multiple strong reasoning models to generate answers independently, followed by majority voting to filter out unsolvable questions and those with excessive answer distribution divergence. For CoT trace selection, we calculate the TF-IDF importance scores and then manually determine the threshold. We reduced the proportion of homogeneous reasoning traces to enrich the core set with diverse key reasoning patterns.
> 3. **We enhance scalability by training specialized annotators to replace direct annotations from strong models.** Within the core set construction process, we have designed a stratified sampling system that combines difficulty and question type annotators to select high-difficulty candidate questions, ensuring diversity in question types. For pattern chain annotation, we initially use strong model annotations *without restricting specific pattern categories*. However, the collected pattern chain annotation data can be used to train lightweight specialized annotation models to reduce costs.
>
> [1] QuRating: selecting high-quality data for training language models, Wettig et al. ICML 2024.
>
> >### `W2&Q1: The similarity computation (weighted DTW combined with the Hungarian algorithm) could be computationally expensive when applied to very large data pools, raising concerns about efficiency. / Could the authors provide an analysis of the computational cost of their similarity matching pipeline at scale?`
>
> Thanks for your insightful question regarding the computational cost. **Both the weighted DTW and the Hungarian algorithm consume limited CPU resources, which are significantly lower than the GPU resources required for model training.** Specifically, the CPU cost is primarily composed of two parts:
> 1. The Weighted DTW Calculation: For example, in practical scenarios with 10,000 core data points, 5,000,000 source data points, and each core selecting 50 source data points, it requires approximately 2,500 CPU days to calculate the distance of pattern chains and entropy chains.
> 2. The Hungarian Algorithm: Optimized through random division into 10 buckets, it requires approximately 1.9 hours. This bucket method ensures optimality within buckets, consistent distribution, and guarantees no overlap.
>
> These costs are considerably lower than the costs associated with model mid-training, and the performance improvement gained is significantly greater than the computational investment, resulting in a favorable return on investment (ROI). Therefore, the computational expense is acceptable.

---

> ### Author Response · Authors · 2025-11-21
> **Response to Reviewer 5gwE (part 2/3)**
>
> >### `W3&Q2&Q3: The reliance on Chinese‑expressed reasoning patterns for distance computation lacks empirical justification, and the paper does not clearly explain how consistency and quality of the large set of abstracted patterns are maintained at scale. / The paper relies on reasoning patterns expressed in Chinese to compute distances between pattern chains. However, it is unclear whether there is empirical evidence demonstrating that this language‑specific representation is optimal or robust. Could the authors provide experiments or ablations to justify this choice? / Moreover, when abstracting a large number of reasoning patterns from massive datasets, how do the authors ensure that the extracted patterns remain consistent, controlled, and aligned with the intended definition of “high‑value” reasoning steps? What mechanisms are in place to prevent noise or drift in the pattern inventory?`
>
> Thanks for prompting us to elaborate on the pattern expression and control of consistency and quality. We conduct comparative experiments and demonstrate that **Chinese pattern representation significantly enhances the distinction of unrelated pattern pairs under the character-level n-gram method, which is beneficial for the consistency and controllability of large-scale extraction of reasoning patterns.**
>
> We sample pattern pairs from the CoTP dataset, among which we manually select unrelated pattern pairs that do not share core mathematical concept words. For example, "Problem Understanding and Information Extraction" versus "Trigonometric Identity Transformation" are unrelated because one pertains to problem understanding while the other involves a specific mathematical technique. Conversely, "Verification and Confirmation" and "Verification and Adjustment" are related as they both pertain to verification. We evaluate their similarity distributions, including related and unrelated ones, under both Chinese and English representations using the n-gram method. Our analysis reveals that **Chinese pattern representation provides superior discriminability under the character-level n-gram method**. The key findings are:
>
> **Table 1: Most pattern pairs in English are identified as either moderately similar or highly similar, making it difficult to differentiate between unrelated patterns, while Chinese offers a superior ability to distinguish between patterns.**
>
> | Similarity Range | Chinese | English |
> |-----------|------|------|
> | Low (<0.3) | 88.2% | 1.4% |
> | Medium  (0.3-0.7) | 11.2% | 83.2% |
> | High (>0.7) | 0.6% | 15.4% |
>
> **Table 2: For the average similarity of unrelated pattern pairs, Chinese can score these pairs with low similarity, whereas English tend to misjudge them as similar.**
>
> | Language | Average Similarity |
> |------|-----------|
> | Chinese | 0.035 |
> | English | 0.541 |
>
> **Table 3: Typical Misjudgment Cases (High Similarity in English but Low in Chinese)**
>
> | Chinese Pattern Pair | Chinese Similarity | English Pattern Pair | English Similarity | Root Cause |
> |----------|-----------|----------|-----------|---------|
> | 问题理解与信息提取 vs 三角恒等变换 | 0.000 | Problem Understanding and Information Extraction vs Trigonometric Identity Transformation | 0.762 | High repetition of suffixes "-tion", "-ation" |
> | 数学建模与方程构建 vs 逻辑排除法 | 0.000 | Mathematical Modeling and Equation Construction vs Logical Elimination Method | 0.760 | Overlapping character-level suffixes |
> | 验证与修正 vs 三角恒等变换 | 0.000 | Verification and Correction vs Trigonometric Identity Transformation | 0.758 | Common suffix interference |
>
> These results demonstrate that under the character-level n-gram method, Chinese pattern representation effectively distinguishes unrelated patterns. It indicates that **even if there is slight noise or drift between patterns in Chinese, such as a small amount of character interference, this method is essentially insensitive to these variants and can still identify semantically similar patterns without the need for explicit explicit deduplication and merging operations**.

---

> ### Author Response · Authors · 2025-11-21
> **Response to Reviewer 5gwE (part 3/3)**
>
> >### `Q4: The proposed CoTP framework shows consistent and significant improvements on mathematical reasoning benchmarks, yet it does not yield comparable gains on general‑domain tasks, sometimes even underperforming other baselines. Could the authors clarify why the method appears highly effective in math reasoning but does not extend to broader general benchmarks? Are there inherent limitations in the design of CoTP that make it domain‑specific?`
>
> Thanks for your question regarding the evaluation of the CoTP framework. Please allow us to clarify our primary objective: **our goal is to significantly expand the reasoning potential of the foundation model for complex mathematical reasoning tasks while maintaining general performance without degradation, even when the concentration of general data is reduced.**
>
> **The gaps observed in general capabilities are primarily due to the data pool being a mix of multiple sources with varying performance.** Our data pool, LongCoTPool, comprises three sources: OpenR1-Math, AM-Distilled, and BoostQA, with BoostQA showing relatively weaker performance on general benchmarks. Given that CoTP is derived by selection from the LongCoTPool, its performance on general benchmarks might be weaker compared to pure OpenR1-Math or AM-Distilled, as indicated in Table 6. However, this does not affect our objective and the main conclusions of our experiments.
>
> **Our method is applicable to diverse domains, including complex mathematical reasoning tasks.** To demonstrate the generalizability of our pattern extraction and importance scoring methodology, we applied the same framework to annotated CoT data from other STEM domains beyond mathematics. The following table presents representative high-importance and low-importance patterns extracted from different STEM domains, demonstrating that our methodology effectively captures domain-specific reasoning characteristics:
> | Domain           | Importance | Pattern Name                      | TF-IDF Score |
> |------------------|------------|-----------------------------------|--------------|
> | Physics          | high       | Periodic Structure Analysis       | 0.915        |
> | Physics          | high       | Electron Conservation Reasoning   | 0.911        |
> | Physics          | high       | Kirchhoff's Law Application       | 0.948        |
> | Physics          | low        | Thermodynamic Data Retrieval      | 0.134        |
> | Physics          | low        | Mechanical Balance Analysis       | 0.130        |
> | Physics          | low        | Unit Conversion and Scaling       | 0.122        |
> | Chemistry        | high       | Mass Ratio Derivation             | 0.932        |
> | Chemistry        | high       | Solution Concentration Backtracking | 0.905      |
> | Chemistry        | low        | Ion Reaction Analysis             | 0.199        |
> | Chemistry        | low        | Chemical Reaction Backtracking    | 0.198        |
> | Biology          | high       | Tissue Property Analysis          | 0.918        |
> | Biology          | high       | Dynamic Equilibrium Analysis      | 0.888        |
> | Biology          | low        | Conclusion Synthesis              | 0.183        |
> | Biology          | low        | Comparative Reasoning             | 0.161        |
> | Computer Science | high       | Algorithm Complexity Analysis     | 0.927        |
> | Computer Science | high       | Dynamic Programming Modeling      | 0.887        |
> | Computer Science | low        | Recursive Function Analysis       | 0.198        |
> | Computer Science | low        | Subtree Analysis                  | 0.181        |
> | ...              | ...        | ...                               | ...          |
>
> These results demonstrate that our pattern extraction and TF-IDF-based importance scoring methodology successfully generalizes across diverse STEM domains, effectively identifying domain-specific high-value reasoning patterns beyond mathematical reasoning tasks.
>
> ### `Conclusion`
> We sincerely appreciate your constructive suggestions on the core set construction, the computational cost, the expression of reasoning patterns, and the evaluation, and we will clarify the relevant content clearly in the revision.

---

> ### Author Response · Authors · 2025-11-27
>
> Dear Reviewer 5gwE,
>
> We would like to thank you once again for your helpful feedback and support. We would greatly appreciate it if you could let us know whether our response addresses your questions.
>
> We sincerely look forward to your reply.
>
> Thank you!

---

### Official Review · Reviewer_1xzE · 2025-10-29

**Soundness:** 2
**Presentation:** 3
**Contribution:** 2
**Rating:** 4
**Confidence:** 4

**Summary:**

The paper discusses which kind of chain of thought data expand the reasoning potential of a foundation model when used during mid-training and how they should be selected?
The authors annotate CoT chains with an atomic reasoning pattern and token wise entropy and weight patterns using TF-IDF to score them with a weighted DTW and use Hungarian algo to match each core instance with the closest candidate data point. With this, they use a pre-trained 85A6B base model and do a mid training with 10B tokens on selected COTP data. They achieved better performance on mathematical datasets. They further perform ablations to justify the importance of entropy based token selection.

**Strengths:**

- The paper uses the reasoning potential of a language model to motivate a data selection process. This is very interesting and justifies the need for such a method to be used during mid-training.
- The paper is clearly presented and each step in the process is well specified with neat plots and algorithms wherever needed. I would say the paper is easy to read and well structured.
- The results looks promising with gains on mathematical datasets which justifies the need for selecting high quality CoT chains.

**Weaknesses:**

- I think the major drawback in the paper is the experimentation limited to a single model on a single domain. Moreover, the “atomic patterns” are produced by another external model like Deepseek and the prompt in Chinese (as provided in the Appendix) seems quite restricted to work on a specific kind of model with a specific kind of dataset. I am not sure how generalizable this approach can be based on the prompt used, model used and the dataset. Also, it might work because of the similarity of the teacher model used (Deepseek) alongside the student model 85A6B and might not work with different combinations. Also, the RL used is limited to one RL algo of GSPO and model limited to MoEs.
- Also, some more ablations are needed to justify all the steps in the process used. For example, a sweep of lambda can be used to show the usefulness of entropy, Hungarian also seems a bit extra and what if we replace it with simple kNN, comparing COTP with a random subset of longCoT given its performance, and so on. This seems that the complexity of each step and all the choices made needs to be studied in more detail.
- Finally, the use of geometric trails is missing from the constraint or the selection process for COTP it seems. Maybe I am missing something but the distance algo is not bounded by the geometric trials in any way and I am wondering if there is any use for geometric trials in itself? Is it more theoretical ?

**Questions:**

1. I want the authors to shed more light on the sensitivity of lambda? What happens when we change it?
2. Also, I am not sure if Hungarian algo is truly needed here? What if we use kNN and why it will not work?
3. Some more ablations in terms of using a random subset of longCoT vs COTP is needed to understand the true usefulness of COTP.
4. Any comment on the generalization of the approach beyond Maths, beyond this specific MoE models (maybe to a dense model), beyond mathematical datasets, using a different model to annotate instead of Deepseek, non chinese prompts and so on.
5. I am also wondering if the contamination test was only done for MathArena or for all the mathematical datasets?
6. Can the approach be extended to other RL recipes beyond GSPO?
7. I am also wondering if mid-training is the best place for such a recipe? I have seen some papers like SIKeD [1]  where they choose datasets during the fine-tuning stage (so in this case, post-training probably) and that improves the performance of the model. I am wondering if you want to comment on that ?

[1] SIKeD: https://aclanthology.org/2025.findings-acl.513/

---

> ### Author Response · Authors · 2025-11-21
> **Response to Reviewer 1xzE (part 1/5)**
>
> Thanks for taking the time to carefully review our paper and to provide thoughtful feedback and questions. Please find our responses to your questions below.
>
> >### `W1&Q4&Q5: I think the major drawback in the paper is the experimentation limited to a single model on a single domain. ... / Any comment on the generalization of the approach beyond Maths, beyond this specific MoE models (maybe to a dense model), beyond mathematical datasets, using a different model to annotate instead of Deepseek, non chinese prompts and so on. / Can the approach be extended to other RL recipes beyond GSPO?`
>
> Thanks for your feedback regarding the generalization of our research. **Our method demonstrates strong generalization as a model architecture-agnostic framework for enhancing reasoning capabilities across diverse domains.** Below we address your concerns point by point.
> ### **(1) Our method is applicable to diverse domains, including complex mathematical reasoning tasks.**
> To demonstrate the generalizability of our pattern extraction and importance scoring methodology, we applied the same framework to annotated CoT data from other STEM domains beyond mathematics. The following table presents representative high-importance and low-importance patterns extracted from different STEM domains, demonstrating that our methodology effectively captures domain-specific reasoning characteristics:
> | Domain           | Importance | Pattern Name                      | TF-IDF Score |
> |------------------|------------|-----------------------------------|--------------|
> | Physics          | high       | Periodic Structure Analysis       | 0.915        |
> | Physics          | high       | Electron Conservation Reasoning   | 0.911        |
> | Physics          | high       | Kirchhoff's Law Application       | 0.948        |
> | Physics          | low        | Thermodynamic Data Retrieval      | 0.134        |
> | Physics          | low        | Mechanical Balance Analysis       | 0.130        |
> | Physics          | low        | Unit Conversion and Scaling       | 0.122        |
> | Chemistry        | high       | Mass Ratio Derivation             | 0.932        |
> | Chemistry        | high       | Solution Concentration Backtracking | 0.905      |
> | Chemistry        | low        | Ion Reaction Analysis             | 0.199        |
> | Chemistry        | low        | Chemical Reaction Backtracking    | 0.198        |
> | Biology          | high       | Tissue Property Analysis          | 0.918        |
> | Biology          | high       | Dynamic Equilibrium Analysis      | 0.888        |
> | Biology          | low        | Conclusion Synthesis              | 0.183        |
> | Biology          | low        | Comparative Reasoning             | 0.161        |
> | Computer Science | high       | Algorithm Complexity Analysis     | 0.927        |
> | Computer Science | high       | Dynamic Programming Modeling      | 0.887        |
> | Computer Science | low        | Recursive Function Analysis       | 0.198        |
> | Computer Science | low        | Subtree Analysis                  | 0.181        |
> | ...              | ...        | ...                               | ...          |
>
> These results demonstrate that our pattern extraction and TF-IDF-based importance scoring methodology successfully generalizes across diverse STEM domains, effectively identifying domain-specific high-value reasoning patterns beyond mathematical reasoning tasks. It is important to clarify that **the patterns we extract are represented in Chinese rather than the prompt itself, which can relatively provide better discriminability under the character-level n-gram method**.
>
> **The focus on complex mathematical reasoning in this study was motivated by the following considerations:**
> 1. Research on large reasoning models has emerged as a hotspot within the field of LLMs, with various reasoning models (_e.g._, DeepSeek-R1) gaining significant attention. This paper aims to contribute an effective method to expand the reasoning potential in foundation models.
> 2. Complex mathematical reasoning tasks impose stricter demands on CoT rigor and step dependency compared to other domains, which can more sensitively reflect differences in various frameworks.
> 3. As shown in Table 6, existing models perform relatively well on commonsense/logical tasks (_e.g._, BBH, DROP) but exhibit significant weaknesses (up to 20% gaps, Table 1) on complex mathematical reasoning tasks (_e.g._, AIME 2025/2024), underscoring their discriminability.

---

> ### Author Response · Authors · 2025-11-21
> **Response to Reviewer 1xzE (part 2/5)**
>
> ### **(2) Our method is agnostic to model architecture.**
> Specifically:
> 1. **Our method is fundamentally a data selection strategy targeting high-value reasoning data, which is agnostic to model architecture.** As long as the model possesses sufficient reasoning capacity, the selected dataset consistently improves performance, whether applied to MoE or dense architectures.
> 2. **Complex mathematical reasoning tasks demand high model reasoning capacity, typically requiring large-scale parameters，and MoE offers a balance between parameter scale and training cost.** Smaller models (_e.g._, 1B/3B parameters) exhibit limited disparity on such complex benchmarks, making it difficult to validate the advantages of data selection strategies. To ensure statistically significant results, we opted for large-scale models. MoE architectures allow scaling model size while significantly reducing computational and training costs compared to dense models, offering greater experimental feasibility and resource efficiency. Thus, MoE became our primary choice.
> 3. **The current mainstream high-performance LLMs predominantly adopt MoE architectures, ensuring the relevance and broad applicability of our conclusions.** Our experiments are conducted on an 85B-parameter MoE model. We anticipate that our method could be transferred to dense models with comparable parameter scales, yielding similar improvements when computational resources permit. In future work, we plan to further validate the generalization of our method across diverse architectures and model scales.
>
> ### **(3) Our method focuses on expanding reasoning potential in foundation models, with RL serving only as downstream validation.**
> Specifically:
> 1. **The proposed method is utilized to select high-value reasoning data during the mid-training stage, fundamentally decoupled from RL algorithms.** Its core contribution lies in demonstrating how the selected high-quality data expands model reasoning potential, rather than providing temporary gains, which is independent of the downstream RL algorithms.
> 2. **GSPO was selected primarily for its architectural compatibility.** As noted in GSPO [1], its sequence-level optimization mechanism better stabilizes MoE RL training compared to token-level algorithms like GRPO, but this doesn't mean the method is limited to the GSPO algorithm. We believe that as long as the upstream reasoning potential is indeed expanded, any effective RL algorithm chosen downstream (_e.g._ GRPO) can be effective and achieve similar improvements.
>
> [1] Group sequence policy optimization, Zheng et al. arXiv preprint arXiv:2507.18071 (2025).

---

> ### Author Response · Authors · 2025-11-21
> **Response to Reviewer 1xzE (part 3/5)**
>
> >### `W2&Q1&Q2&Q3: Also, some more ablations are needed to justify all the steps in the process used. ... / I want the authors to shed more light on the sensitivity of lambda? What happens when we change it? / Also, I am not sure if Hungarian algo is truly needed here? What if we use kNN and why it will not work? / Some more ablations in terms of using a random subset of longCoT vs COTP is needed to understand the true usefulness of COTP.`
>
> We provide additional clarifications on ablation studies:
> ### **(1) Our ablations demonstrate that neither pattern chain selection nor entropy chain selection alone outperforms the hybrid method.**
> The lambda coefficient balances these two selection strategies. When lambda=1, the algorithm relies solely on pattern chain selection, prioritizing the abstraction of reasoning patterns in the trajectory. When lambda=0, it switches entirely to entropy chain selection, focusing on fine-grained token-level features in the reasoning steps. Our experiments demonstrate that balancing both strategies can yield better performance in complex reasoning scenarios.
> ### **(2) Compared to kNN, the Hungarian algorithm is suited to our framework for ensuring theoretical completeness and rigor, as it guarantees unique matching, eliminates hyperparameter tuning, and maintains data distribution consistency.**
> Specifically:
> 1. The Hungarian algorithm ensures that each source data is selected at most once, achieving strict one-to-one matching. In our experiments, it achieved 100% unique coverage and 0% overlap, guaranteeing each core point matches its optimal source counterpart. Systematic comparisons on CoTP datasets show naive kNN (k∈{1,5,10}) causes severe overlap: at k=1, unique coverage was only 84% with 16.0% overlap; at k=5, coverage dropped to 62% with 38.0% overlap; at k=10, coverage further declined to 44% with 56.0% overlap. This occurs because popular source data may resemble multiple core samples, creating competition that leaves some core points without optimal matches.
> 2. The Hungarian algorithm requires no hyperparameters (_e.g._, capacity control), ensuring well-defined, reproducible results.
> 3. By preventing overlap, it maintains high distributional alignment between selected training data and the core set (Figure 6), whereas kNN's overlap disrupts balance—over-selected popular samples may cluster in certain problem types, skewing the distribution.
>
> We acknowledge kNN's computational efficiency merits for large-scale systems, and the following heuristic strategies could partially mitigate the above issues:
> 1. Capacity Control: Restrict each source sample to be assigned at most once, preventing overuse of a few _popular_ samples.
> 2. Fallback Strategy: When the nearest neighbour of a core point has already been assigned, fall back to the next closest unassigned neighbour (2nd, 3rd, …) until a valid match is found.
> 3. Local Refinement: Perform lightweight local optimization or greedy swapping on top of an initial kNN matching to reduce conflicts and improve the total matching quality.
> Such hyperparameter-dependent solutions may work pragmatically but lack theoretical guarantees and introduce implementation ambiguity. For theoretical rigor, we adopted the standard Hungarian algorithm as our primary method.
> ### **(3) We have conducted comparative experiments between random subsets of LongCoT and CoTP, with results demonstrating CoTP's superior reasoning performance.**
> In our experimental setup, we established LongCoTPool as a comparative baseline. Since the size of the training set is fixed, it is essentially constructed from a random subset of the data pool (detailed in Section 3.1). The results show that CoTP improves by 6.39% over LongCoTPool, with a remarkable 9.58% enhancement on AIME 2025 & 2024 (line 288).

---

> ### Author Response · Authors · 2025-11-21
> **Response to Reviewer 1xzE (part 4/5)**
>
> >### `W3: Finally, the use of geometric trails is missing from the constraint or the selection process for COTP it seems. Maybe I am missing something but the distance algo is not bounded by the geometric trials in any way and I am wondering if there is any use for geometric trials in itself? Is it more theoretical ?`
>
> **The core contribution of geometric trials lies in establishing a quantitative evaluation framework for model potential, which guides the construction of the core set.** Specifically:
> 1. **Geometric trials rigorously define model reasoning potential by statistically modeling the _first-passage time (K)_**, the number of independent attempts required to solve a given question. A smaller K indicates higher potential. This theory demonstrates that exposing the model to diverse effective reasoning paradigms, rather than random rollouts, significantly improves the success rate per attempt. This insight directly guides our core set construction, prioritizing samples that exhibit high reasoning efficiency, namely low-K characteristics.
> 2. **Geometric trials and the distance algorithm form a synergistic theory-practice relationship.** While the formalization of the distance algorithm is not directly constrained by geometric trials, the two collaborate as follows: The potential metrics from geometric trials define the _high-value reasoning diversity_ standard that the core set should satisfy, and the distance algorithm efficiently selects samples from the source pool that align with the core set distribution through spatial metric computations, thereby preserving the theoretically desired high-potential properties.
>
> >### `Q5: I am also wondering if the contamination test was only done for MathArena or for all the mathematical datasets?`
>
> Our study strictly adheres to comprehensive evaluation principles, and **the contamination test was conducted across all benchmark datasets (including both general benchmarks and complex mathematical reasoning benchmarks)** (line 229), to ensure the reliability of our experimental conclusions across diverse task scenarios.

---

> ### Author Response · Authors · 2025-11-21
> **Response to Reviewer 1xzE (part 5/5)**
>
> >### `Q7: I am also wondering if mid-training is the best place for such a recipe? I have seen some papers like SIKeD where they choose datasets during the fine-tuning stage (so in this case, post-training probably) and that improves the performance of the model. I am wondering if you want to comment on that ?`
>
> Thanks for raising this insightful discussion regarding the best place for the recipe. **For the complex mathematical reasoning tasks that our research focuses on, introducing such high-value reasoning data during mid-training strikes the best balance between enhancing reasoning capabilities and preserving generalization performance.** It is grounded in several considerations:
> 1. **The difficulty of the task significantly influences the appropriate stage for data intervention.** In conventional mathematical tasks like GSM8k, which are addressed in studies such as SIKeD, foundation models pre-trained on general corpora already possess substantial potential. In these scenarios, fine-tuning with a modest amount of task-specific data suffices to boost performance, which is why studies like SIKeD opt for data selection during the fine-tuning stage.
> 2. **The mid-training stage proves more suitable for data scaling [1, 2, 3].** For complex mathematical reasoning tasks, general pre-training data lacks sufficient reasoning-intensive samples, such as high-difficulty long-CoT sequences, which are necessary for models to acquire the core reasoning paradigms required to solve such tasks. In practice, we often synthesize high-difficulty reasoning data at a scale of billions (_e.g._, 50B data) to enhance the model's reasoning abilities. If this data is directly added to the pre-training stage, its concentration can be extremely low (_e.g._, 50B data blended with 10T pre-training data results in a mere 0.5% concentration), making it difficult to achieve the desired impact. Conversely, utilizing all this data in the fine-tuning stage could potentially impair the model's generalization ability and lead to overfitting on specific reasoning tasks, given that 50B data corresponds to approximately 2 million long-CoT sequences.
> 3. **Recent research has demonstrated that the mid-training stage is highly suitable for enhancing reasoning ability [4].** By strategically combining high-quality general data with reasoning-intensive data, the concentration can be significantly increased (_e.g._, 50B data out of total 200B mid-training data results in a 25% concentration). This not only expands the model's reasoning potential but also maintains robust generalization capabilities. Subsequently, supplementing with a small amount of fine-tuning data can fully activate the model's reasoning abilities and further improve its performance on complex reasoning benchmarks. Additionally, this strategy aligns well with the subsequent RL stage, enhancing the upper limit of reasoning performance.
>
> [1] Large-Scale Diverse Synthesis for Mid-Training, Zhang et al. arXiv preprint arXiv:2508.01326 (2025).
>
> [2] Midtraining Bridges Pretraining and Posttraining Distributions, Liu et al. arXiv preprint arXiv:2510.14865 (2025).
>
> [3] A Survey on LLM Mid-training, Tu et al. arXiv preprint arXiv:2510.23081 (2025).
>
> [4] Octothinker: Mid-training incentivizes reinforcement learning scaling, Wang et al. arXiv preprint arXiv:2506.20512 (2025).
>
> ### `Conclusion`
> We sincerely appreciate your detailed constructive suggestions on generalization, ablations, geometric trials, contamination test, and the stage for our method. We hope our responses have addressed your concerns and strengthened confidence in our paper, and we will add relevant details in the revision.

---

> ### Author Response · Authors · 2025-11-27
>
> Dear Reviewer 1xzE,
>
> We would like to thank you once again for your helpful feedback and would greatly appreciate it if you could let us know whether our response addresses your concerns.
>
> We respectfully invite you to reconsider our work favorably. **We propose a novel CoTP framework to select data rich in high-value reasoning patterns, which greatly expands the reasoning potential of the foundation model (+9.58% on AIME) and significantly raises the upper bound of downstream RL performance (+7.81%).**
>
> We sincerely look forward to your reply.
>
> Thank you!

---

### Official Review · Reviewer_zwnc · 2025-10-31

**Soundness:** 4
**Presentation:** 3
**Contribution:** 4
**Rating:** 8
**Confidence:** 3

**Summary:**

This paper introduces CoTP, a data selection framework that works by expanding the dataset with the COT patterns to increase what the authors define as the model's reasoning potential. The model reasoning potential is defined as the inverse of the number of independent attempts required to correctly answer a question. Using CoTP data the authors are able to improve an MoE model by 9.58% on AIME dataset.

**Strengths:**

- **Novelty**: The paper presents an original and clearly articulated approach to expanding reasoning potential in foundation models. Both the methodological design and the underlying problem formulation demonstrate genuine novelty.

- **Elegant Solution**: The combination of core-set construction with a dual-granularity selection strategy represents an elegant and well-motivated solution that advances beyond conventional Chain-of-Thought augmentation methods.

- **Theoretical Rigor**: The framework is supported by a strong theoretical foundation and is further reinforced through detailed empirical analysis, illustrating both conceptual robustness and practical effectiveness.

- **Comprehensive Empirical Validation**: The work provides extensive experimental results and ablation studies that empirically validate the proposed theoretical framework, showing consistent improvements across mathematical reasoning benchmarks.

- **Reproducibility**: Code was added as supplementary materials, which enhances the transparency and reproducibility of the work.

**Weaknesses:**

- **Limited Evaluation Scope**: The experiments focus exclusively on mathematical reasoning tasks, leaving uncertainty about whether the proposed framework generalizes to broader domains such as logical, commonsense, or multimodal reasoning.
- **Dependence on DeepSeek-R1**: The framework relies heavily on DeepSeek-R1 for generating Chain-of-Thought (CoT) traces, which is computationally expensive. It would be valuable to examine whether similar improvements can be achieved using smaller or alternative base models.
- **Limited Cross-Model Evaluation**: The study evaluates the method on only a single MoE (mixture-of-experts) model configuration, providing little evidence of robustness across different architectures or model families. Broader testing across diverse model types would strengthen the general applicability claims.

**Questions:**

The paper mentions a manual screening process for selecting high-value reasoning traces (line 165). Could the authors clarify what this process entails? Specifically, how were examples chosen, and what criteria or safeguards were used to minimize potential bias or subjectivity in the selection?

---

> ### Author Response · Authors · 2025-11-21
> **Response to Reviewer zwnc (part 1/3)**
>
> Thank you for your feedback and support of our work. Please find our responses to your questions below.
>
> >### `W1: Limited Evaluation Scope: The experiments focus exclusively on mathematical reasoning tasks, leaving uncertainty about whether the proposed framework generalizes to broader domains such as logical, commonsense, or multimodal reasoning.`
>
> **Our method is applicable to diverse domains, including complex mathematical reasoning tasks**, as supported by the following aspects:
> 1. **Architecture Design**: The data selection algorithm (Section 2) is domain-agnostic and theoretically applicable to any scenario decomposable into atomic reasoning patterns (_e.g._, STEM fields).
> 2. **Preliminary Validation**: To demonstrate the generalizability of our pattern extraction and importance scoring methodology, we applied the same framework to annotated CoT data from other STEM domains beyond mathematics. The following table presents representative high-importance and low-importance patterns extracted from different STEM domains, demonstrating that our methodology effectively captures domain-specific reasoning characteristics:
> | Domain           | Importance | Pattern Name                      | TF-IDF Score |
> |------------------|------------|-----------------------------------|--------------|
> | Physics          | high       | Periodic Structure Analysis       | 0.915        |
> | Physics          | high       | Electron Conservation Reasoning   | 0.911        |
> | Physics          | high       | Kirchhoff's Law Application       | 0.948        |
> | Physics          | low        | Thermodynamic Data Retrieval      | 0.134        |
> | Physics          | low        | Mechanical Balance Analysis       | 0.130        |
> | Physics          | low        | Unit Conversion and Scaling       | 0.122        |
> | Chemistry        | high       | Mass Ratio Derivation             | 0.932        |
> | Chemistry        | high       | Solution Concentration Backtracking | 0.905      |
> | Chemistry        | low        | Ion Reaction Analysis             | 0.199        |
> | Chemistry        | low        | Chemical Reaction Backtracking    | 0.198        |
> | Biology          | high       | Tissue Property Analysis          | 0.918        |
> | Biology          | high       | Dynamic Equilibrium Analysis      | 0.888        |
> | Biology          | low        | Conclusion Synthesis              | 0.183        |
> | Biology          | low        | Comparative Reasoning             | 0.161        |
> | Computer Science | high       | Algorithm Complexity Analysis     | 0.927        |
> | Computer Science | high       | Dynamic Programming Modeling      | 0.887        |
> | Computer Science | low        | Recursive Function Analysis       | 0.198        |
> | Computer Science | low        | Subtree Analysis                  | 0.181        |
> | ...              | ...        | ...                               | ...          |
>
> These results demonstrate that our pattern extraction and TF-IDF-based importance scoring methodology successfully generalizes across diverse STEM domains, effectively identifying domain-specific high-value reasoning patterns beyond mathematical reasoning tasks.
>
> **The focus on complex mathematical reasoning in this study was motivated by the following considerations:**
> 1. Research on large reasoning models has emerged as a hotspot within the field of LLMs, with various reasoning models (_e.g._, DeepSeek-R1) gaining significant attention. This paper aims to contribute an effective method to expand the reasoning potential in foundation models.
> 2. Complex mathematical reasoning tasks impose stricter demands on CoT rigor and step dependency compared to other domains, which can more sensitively reflect differences in various frameworks.
> 3. As shown in Table 6, existing models perform relatively well on commonsense/logical tasks (_e.g._, BBH, DROP) but exhibit significant weaknesses (up to 20% gaps, Table 1) on complex mathematical reasoning tasks (_e.g._, AIME 2025/2024), underscoring their discriminability.

---

> ### Author Response · Authors · 2025-11-21
> **Response to Reviewer zwnc (part 2/3)**
>
> >### `W2: Dependence on DeepSeek-R1: The framework relies heavily on DeepSeek-R1 for generating Chain-of-Thought (CoT) traces, which is computationally expensive. It would be valuable to examine whether similar improvements can be achieved using smaller or alternative base models.`
>
> Thanks for your feedback. **The use of smaller yet capable reasoning models is indeed feasible.** Recent studies [1] demonstrate that certain models with small parameters (_e.g._, DeepSeek-R1-Distill-Qwen) can achieve comparable reasoning performance to larger models through optimizations like knowledge distillation and architectural improvements. **Our choice of DeepSeek-R1 as the generative model is primarily based on the following considerations:**
> 1. **It enables us to align with baselines and control experimental variables.** All long-CoT datasets in our baseline studies were generated using DeepSeek-R1. Maintaining consistency in the generative model helps isolate variables, thereby more clearly demonstrating the improvement effects of our proposed framework itself.
> 2. **The reasoning performance of DeepSeek-R1 meets the experimental requirements.** As a mainstream strong reasoning model in the industry, DeepSeek-R1 possesses distinctive and effective reasoning paradigms capable of generating diverse, learnable long-CoT traces.
>
> In our experiments, to ensure the completeness and rigor of the conclusions, including a fair comparison with the baseline conditions, we chose DeepSeek-R1 as the generative model.
>
> [1] Deepseek-r1: Incentivizing reasoning capability in llms via reinforcement learning, Guo et al. arXiv preprint arXiv:2501.12948 (2025).
>
> >### `W3: Limited Cross-Model Evaluation: The study evaluates the method on only a single MoE (mixture-of-experts) model configuration, providing little evidence of robustness across different architectures or model families. Broader testing across diverse model types would strengthen the general applicability claims.`
>
> **Our method is fundamentally a data selection strategy targeting high-value reasoning data, which is agnostic to model architecture.** Specifically:
> 1. **As long as the model possesses sufficient reasoning capacity, the selected dataset consistently improves performance**, whether applied to MoE or dense architectures.
> 2. **Complex mathematical reasoning tasks demand high model reasoning capacity, typically requiring large-scale parameters，and MoE offers a balance between parameter scale and training cost.** Smaller models (_e.g._, 1B/3B parameters) exhibit limited disparity on such complex benchmarks, making it difficult to validate the advantages of data selection strategies. To ensure statistically significant results, we opted for large-scale models. MoE architectures allow scaling model size while significantly reducing computational and training costs compared to dense models, offering greater experimental feasibility and resource efficiency. Thus, MoE became our primary choice.
> 3. **The current mainstream high-performance LLMs predominantly adopt MoE architectures, ensuring the relevance and broad applicability of our conclusions.** Our experiments are conducted on an 85B-parameter MoE model. We anticipate that our method could be transferred to dense models with comparable parameter scales, yielding similar improvements when computational resources permit.
>
> In summary, our method is inherently data-centric and agnostic to model architecture. Our choice of MoE was based on a pragmatic trade-off between resource constraints and performance requirements for complex reasoning tasks, aligning with the prevailing trend in LLM development. In future work, we plan to further validate the generalization of our method across diverse architectures and model scales.

---

> ### Author Response · Authors · 2025-11-21
> **Response to Reviewer zwnc (part 3/3)**
>
> >### `Q1: The paper mentions a manual screening process for selecting high-value reasoning traces (line 165). Could the authors clarify what this process entails? Specifically, how were examples chosen, and what criteria or safeguards were used to minimize potential bias or subjectivity in the selection?`
>
> Thanks for your question regarding our manual screening process. **Our manual screening was conducted with the assistance of statistical non-manual methods to ensure core set quality while minimizing subjectivity through these key measures:**
> 1. During question screening, manual intervention was limited to quality and answer correctness verification. Specifically, we employed multiple strong reasoning models to generate answers independently, followed by majority voting to filter out unsolvable questions and those with excessive answer distribution divergence.
> 2. For CoT trace selection, we calculate the TF-IDF importance scores and then manually determine the threshold. We reduced the proportion of homogeneous reasoning traces to enrich the core set with diverse key reasoning patterns.
>
> ### `Conclusion`
>
> We sincerely appreciate your constructive suggestions on the evaluation, the generative model, and manual screening, which help us to clarify the relevant content. We will include these points in the revision.

---

> ### Comment · Reviewer_zwnc · 2025-11-25
>
> I thank the authors for addressing my concerns. I will keep the score as is.

---

### Official Review · Reviewer_vJQb · 2025-11-03

**Soundness:** 3
**Presentation:** 3
**Contribution:** 3
**Rating:** 8
**Confidence:** 3

**Summary:**

This paper presents a mid-training approach using CoT reasoning patterns. Through a process of selecting long-CoT data that is similar to the core set using similarity measures based on pattern & token entropy. The approach has dramatic empirical gains, advancing SOTA significantly in mathematical reasoning.

**Strengths:**

This paper presents an innovative and detailed approach for a well studied problem.
* **Empirical Results**: Significant gains in the performance of models when using the proposed method of mid training.
* **Innovative approach**: The proposed algorithm is clever, elegant and interesting to read. It is nice to see such a well motivated method work well empirically.
* **Ablations**: The authors provide a very nice set of ablations that help to complete the understanding of the design space of the given approach, especially Table 3.

**Weaknesses:**

Mostly I believe this to be a strong paper, which has interesting and high quality results and methodologies.

There are a lot of nice ablations in the paper, but the results that I think may be missing are: connecting the core set similarity to downstream performance? We see only Figure 6 for LongCotPool and CoTP but not the Table 4 ablations? This will help to prove the claim?

**Questions:**

1. As I understand from line 227, DeepSeek-R1 is the model used for reasoning trace generation. It seems using a variety of datasets is important. Do you think diversity of generative model would be helpful or harmful?

---

> ### Author Response · Authors · 2025-11-21
> **Response to Reviewer vJQb**
>
> Thank you for your feedback and support of our work. Please find our responses to your questions below.
>
> >### `W1: connecting the core set similarity to downstream performance? We see only Figure 6 for LongCotPool and CoTP but not the Table 4 ablations? This will help to prove the claim?`
>
> **We agree that both Figure 6 and Table 4 are instrumental in demonstrating this relationship.** We address your concerns in detail below:
> 1. **Table 4 is designed to validate whether CoTP enables models to learn key reasoning patterns closer to the ideal ones.** Specifically, we consider the model trained on the oracle set as the _ideal_ model, whose generated reasoning traces encompass comprehensive patterns. Our core set serves as an approximation of this oracle. Through the data selection algorithm, we select training data that aligns with the core set distribution. In Table 4, we use the strong reasoning model DeepSeek-R1 as a proxy for the ideal model. By comparing the overlap of reasoning patterns between models trained on different datasets (KnowEdu, LongCoTPool, CoTP) and DeepSeek-R1, we observe that the model trained with CoTP achieves significantly higher overlap in key patterns (detailed in Table 4 and Appendix F.1).
> 2. **The core set is a carefully designed to cover key reasoning patterns, and CoTP selects similar patterns for the experimental model to learn, thereby improving downstream performance.** CoTP not only selects data with distributional similarity (Figure 6) but also learns patterns with higher overlap to the ideal model’s patterns (Table 4). Together, these results explain CoTP’s performance gains on downstream tasks (Table 2), demonstrating the effectiveness of our algorithm and training framework.
>
> >### `Q1: As I understand from line 227, DeepSeek-R1 is the model used for reasoning trace generation. It seems using a variety of datasets is important. Do you think diversity of generative model would be helpful or harmful?`
>
> Thanks for your question regarding model diversity in our research. Concerning the selection of generative models, we fully agree that **employing diverse generative models is crucial for enhancing the variety of long-CoT traces**. In practice, we actively encourage applying our method across multiple reasoning models (_e.g._, Qwen3, GPT5), as this would provide richer references for reasoning patterns.
> **Our choice of DeepSeek-R1 as the generative model is primarily based on the following considerations:**
> 1. **It enables us to align with baselines and control experimental variables.** All long-CoT datasets in our baseline studies were generated using DeepSeek-R1. Maintaining consistency in the generative model helps isolate variables, thereby more clearly demonstrating the improvement effects of our proposed framework itself.
> 2. **The reasoning performance of DeepSeek-R1 meets the experimental requirements.** As a mainstream strong reasoning model in the industry, DeepSeek-R1 possesses distinctive and effective reasoning paradigms capable of generating diverse, learnable long-CoT traces.
>
> ### `Conclusion`
>
> We sincerely appreciate your constructive suggestions on model diversity and the connection between core set similarity and downstream performance. We will incorporate relevant points in the revision.

---

### Author Response · Authors · 2025-12-03
**Summary to Area Chair**

Dear Area Chair,

We sincerely appreciate the time and effort that you and the reviewers have devoted to our submission, as well as the valuable and constructive feedback provided. **For all the concerns and questions raised, we have offered comprehensive responses.** However, due to the unexpected situation with the OpenReview system, we regret that we are unable to engage in further discussion with the reviewers.

In the latest revision, **we have incorporated many of the valuable suggestions that help strengthen our work**, including providing additional details and refining the clarity of our presentation. We kindly invite you to review both the rebuttal and the revised submission.

We would like to take this opportunity to highlight the core contributions of our work. We **theoretically define the reasoning potential of the foundation model for the first time** and conduct a comprehensive investigation from both theoretical and practical perspectives. Specifically, we propose a novel data selection framework, **CoTP**, which selects data rich in high-value reasoning patterns. This framework **significantly expands the reasoning potential of the foundation model (_e.g._, 85A6B MoE), achieving a 9.58% improvement on the challenging AIME 2025&2024 benchmarks and raising the upper bound of downstream RL performance by 7.81%**.

Thank you again for your time and consideration.

---

### Meta-Review · Area_Chair_JaEj · 2026-01-22

**Summary:**

This paper proposes a mid-training data selection framework to increase a model’s “reasoning potential,” operationalized via stochastic decoding success probability (and expected attempts). The approach builds a small core set of high-value long-CoT traces, extracts atomic reasoning-pattern chains and token-entropy chains, and selects training traces from a large pool by minimizing a dual-granularity DTW distance with a global assignment step to reduce overlap. Reviewers agree the method is clearly motivated and technically nontrivial, with strong empirical gains on AIME-style math reasoning and positive downstream RL effects. The decision is driven by the strength and consistency of the improvements and the coherent methodology, tempered by concerns about generality beyond math, reliance on a specific external teacher/annotator, and limited exploration of alternatives/efficiency trade-offs.

**Reviewer Concerns:**

Addressed partially or fully by rebuttal

* Clarifications on the definition/intuition of “reasoning potential,” and why it is relevant to both pass@k and RL performance.

* Justification for the assignment/matching design (Hungarian vs. greedy/kNN) and the role of overlap control; rebuttal explains the failure modes of simpler heuristics.

* Additional implementation clarifications (core-set construction workflow, pattern extraction pipeline) and ablation interpretations.

Still outstanding

+ Generality: evidence remains concentrated in mathematical reasoning and one large MoE setting; transfer to other domains (e.g., logic, code, commonsense, multimodal) and other model families is not demonstrated.

+ Teacher dependence: the pipeline’s reliance on a single external model for pattern annotation/generation remains a key limitation; robustness to teacher choice is not empirically established.

+ Compute/complexity: the cost and scalability of DTW + assignment (and overall selection pipeline) versus simpler baselines is not fully quantified; efficiency/accuracy trade-offs remain somewhat unclear.

+ Potential subjectivity in core-set screening/pattern taxonomy (language/prompting choices) and how this affects reproducibility/transfer.

**Reviewer Scores:**

* R1 (8  ->  8): Likely unchanged; rebuttal strengthens clarity and supports the original positive view but does not add new cross-domain evidence.

* R2 (8  ->  8): Likely unchanged; core claims remain supported, and rebuttal addresses several methodological questions.

* R3 (4  ->  5): Likely to increase slightly due to clarifications on matching/overlap control and rationale for design choices; however, concerns about generality and teacher dependence remain, limiting a larger score change.

---

### Decision · Program_Chairs · 2026-01-26

Accept (Poster)